# Quantifying hydrogen bonding using electrically tunable nanoconfined water

Ziwei Wang [1,2] ✉, Anupam Bhattacharya[1], Mehmet Yagmurcukardes[3], Vasyl Kravets[1], Pablo Díaz-Núñez [1,2], Ciaran Mullan[1], Ivan Timokhin[1,2], Takashi Taniguchi [4], Kenji Watanabe [4], Alexander N. Grigorenko [1], Francois Peeters [5], Kostya S. Novoselov [1,2,6], Qian Yang [1,2] ✉ & Artem Mishchenko [1,2] ✉

Hydrogen bonding plays a crucial role in biology and technology, yet it remains poorly understood and quantified despite its fundamental importance. Traditional models, which describe hydrogen bonds as electrostatic interactions between electropositive hydrogen and electronegative acceptors, fail to quantitatively capture bond strength, directionality, or cooperativity, and cannot predict the properties of complex hydrogen-bonded materials. Here, we introduce a concept of hydrogen bonds as elastic dipoles in an electric field, which captures a wide range of hydrogen bonding phenomena in various water systems. Using gypsum, a hydrogen bond heterostructure with two-dimensional structural crystalline water, we calibrate the hydrogen bond strength through an externally applied electric field. We show that our approach quantifies the strength of hydrogen bonds directly from spectroscopic measurements and reproduces a wide range of key properties of confined water reported in the literature. Using only the stretching vibration frequency of confined water, we can predict hydrogen bond strength, local electric field, O-H bond length, and dipole moment. Our work also introduces hydrogen bond heterostructures – a class of electrically and chemically tunable materials that offer stronger, more directional bonding compared to van der Waals heterostructures, with potential applications in areas such as catalysis, separation, and energy storage.

Hydrogen bonds (HBs) are unique intermolecular interactions that are stronger and more directional than weak and isotropic van der Waals forces, yet weaker and less directional than covalent bonds[1–4]. HBs are typically defined as attractive interactions between a hydrogen atom (H) and an acceptor group (A) in a system represented as D-H⋯A, where H is covalently bonded to an electronegative donor group (D), while H carries a partial positive charge $\delta^+$[5,6]. A key feature of HBs is

their directionality, often with a D-H⋯A angle close to 180°. In water and similar systems, where distance between D and A is typically greater than 2.5 Å, the electrostatic attraction between H ($\delta^+$) and A ($\delta^-$) is believed to be the primary factor contributing to the strength of HBs[7–11].

HBs could significantly alter the properties of residing system. For instance, the formation of HBs in water has a pronounced impact on

[1]Department of Physics and Astronomy, University of Manchester, Manchester, UK. [2]National Graphene Institute, University of Manchester, Manchester, UK. [3]Department of Photonics, Izmir Institute of Technology, Izmir, Turkey. [4]National Institute for Materials Science, Tsukuba, Japan. [5]Department Physics, University of Antwerp, Antwerpen, Belgium. [6]Institute for Functional Intelligent Materials, National University of Singapore, Singapore, Singapore.
✉e-mail: ziwei.wang@manchester.ac.uk; qian.yang@manchester.ac.uk; artem.mishchenko@manchester.ac.uk

the physical properties of its condensed phases−ice. Often, hydrogen-bonded systems form extensive networks of HBs that extend over long ranges and in various directions, as seen in liquid water, ices, hydrogels, proteins, and DNA molecules. The complexity of these networks, along with the presence of interfaces or external electric fields in realistic heterogeneous systems, makes it difficult to analyze them using classic definition of HBs or to predict the properties of such systems[12]. To address these challenges, the electrostatic model of HBs, initially proposed decades ago that based on point charge approximation[13], has been refined with various treatments, such as electron distribution calculations[14] and orbital hybridization theory[15], leading to a more sophisticated description of HBs. Despite advancements in modeling and the introduction of spectroscopic methods, which provide direct information on molecular vibrations of hydrogen-bonded species[16], a practical and efficient experimental approach to quantitatively predict the properties of HBs is still lacking.

Here we propose an approach that conceptualizes an HB as an electric dipole moment $\mathbf{p}$ of the donor-hydrogen (D-H) pair interacting with the electric field $\mathbf{E}_{HB}$ induced by the acceptor A (Fig. 1a). The strength of $\mathbf{E}_{HB}$ at the dipole's position depends on the electronegativity of the acceptor A, and the distance between $\mathbf{E}_{HB}$ and $\mathbf{p}$. Considering that the typical distance between A and the D-H dipole in an HB is often more than twice the length of the dipole, we assume $\mathbf{E}_{HB}$ to be uniform at the dipole's position. Also, the $\mathbf{E}_{HB}$ is the effective electric field from the acceptor, where the influence on it from $\mathbf{p}$ has been embedded in our model. The dipole moment ($\mathbf{p} = q\mathbf{d}$) arises from the separation of the charge $q$ along the D-H bond direction $\mathbf{d}_{D\rightarrow H}$, while the field $\mathbf{E}_{HB}$ generated by the acceptor nearby, A, provides electric potential energy $U_{HB} = -\mathbf{p}\cdot\mathbf{E}_{HB}$. This potential energy, which depends on the magnitude of $\mathbf{p}$, the strength of $\mathbf{E}_{HB}$ at dipole position, and cosine of the angle between them, represents the electrostatic interaction energy of the hydrogen bonding system. It is maximized when the electric field and the dipole are aligned. The field $\mathbf{E}_{HB}$ stretches the D-H dipole, effectively lowering its spring constant from that of an unbounded free D-H bond. When subject to external electric field ($E_{ext}$), the projected component of $E_{ext}$ along the HB direction can further alter the spring constant and dipole moment of D-H dipole, thus changing the overall electrostatic interactions of the hydrogen bonding system. This offers a more universal and precisely quantifiable dipole-in-$E$-field alternative to the abstract chemical bond definition of HBs. Our model reflects both the dominant electrostatic nature of HBs (electric potential energy $U_{HB}$ as a quantitative measure of HB strength) and their directionality (the angle between $\mathbf{p}$ and $\mathbf{E}_{HB}$, Fig. 1b). Moreover, by treating individual HBs as electric fields exerting on separate dipoles, our method allows the overall properties of a system with complex HB networks to be predicted based on the superposition of dipole moments and electric fields. In this work, we introduce the dipole-in-$E$-field approximation for HBs and use

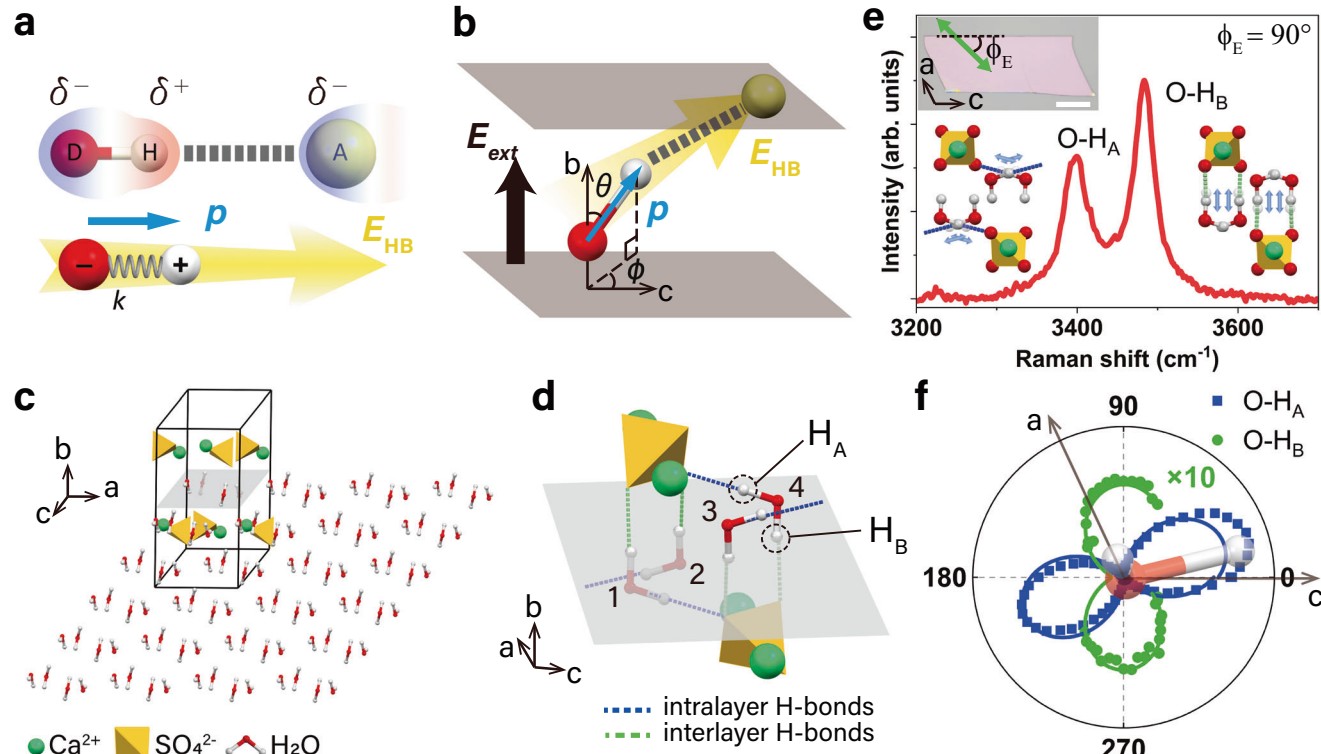

**Fig. 1 | Hydrogen bond model and gypsum system. a** Hydrogen bond models. The upper diagram represents conventional model. Red and green atoms labeled with D and A are negatively charged ($\delta^-$) donor and acceptor, respectively. Hydrogen (H) atom carries partial positive charge ($\delta^+$). The lower diagram represents our dipole-in-$E$-field approach, where the HB is conceptualized as an electrostatic interaction of a dipole oscillator, with spring constant $k$ and dipole moment $p$ (blue arrow), in the local field $E_{HB}$ (yellow arrow) of a negatively charged acceptor. **b** Schematics of the HB in (**a**) when subjected to an external electric field ($E_{ext}$) along the out-of-plane direction. The orientation of the dipole is described by angles $\vartheta$ and $\phi$, which is the angle of D-H bond with respect to in-plane crystalline axis and out-of-plane confining axis. **c** Unit cell of gypsum (in the box) and the extended confined water (oxygen in red, hydrogen in white) plane in the same two-dimensional molecular sheet. The crystal structure is simulated by our density

functional theory study. **d** Zoomed-in sketch of (**c**) showing two types of O-H dipoles with different H atoms, intralayer O-$H_A$ and interlayer O-$H_B$. **e** Raman spectrum of water stretching modes in an exfoliated thick gypsum crystal, assigned to the stretching of O-$H_A$ and O-$H_B$ dipoles. Schematics of stretching vibrations are shown next to the respective peaks as insets. The top inset is an optical micrograph of exfoliated thick gypsum flake on a CaF$_2$ substrate. The polarization of laser is described by $\phi_E$, the angle between laser polarization (green arrow) and gypsum crystalline c-axis (dashed line). Scale bar 30 μm. The spectrum was taken at $\phi_E = 90°$. **f** Angle-resolved Raman intensity of O-$H_A$ (blue squares) and O-$H_B$ stretching (green circles) stretching modes. Solid lines the sinusoidal fittings used to obtain $\phi$ for O-$H_A$ and O-$H_B$ ($\phi$ defined in (**b**)) of water molecules in gypsum. The projection of a water molecule on the gypsum a-c plane is shown in the center as an inset. Source data are provided as a Source Data file.

naturally occurring hydrogen-bonded heterostructure gypsum to quantitatively describe HBs in nanoconfined water.

## Results

### Dipole-in-$E$-field model of hydrogen bonding

Conventionally, the strength of HBs is characterized as a shift in the stretching vibration frequency of the D-H covalent bond: the stronger the HB, the more red-shifted the frequency ($\omega_{D-H}$). Here we consider the HB as D-H dipole in the presence of a local electric field, with the red shift in frequency depending on the local field produced by the acceptor ($\mathbf{E}_{HB}$). In a harmonic first-order approximation, the total potential energy ($U_{osc}$) and electric potential energy ($U_{HB}$) of D-H dipole oscillator in $E$-field are given by:

$$U_{osc} = \frac{1}{2}k(|\mathbf{x}| - |\mathbf{d}|)^2 + U_{HB} \qquad (1)$$

$$U_{HB} = -\mathbf{p} \cdot \mathbf{E} \qquad (2)$$

Here, $k$ is the force constant of the D-H bond, $\mathbf{d}$ and $\mathbf{x}$ are vectors pointing from the donor (at rest) to the equilibrium and instantaneous positions of hydrogen, respectively. The $\mathbf{E}$ field is the effective electric field at the dipole position (see Methods, 'Full form of the dipole-in-$E$-field approximation'), which changes the equilibrium bond length and bond alignment elastically. In the absence of rotation, the problem is simplified to one dimension, with the vectors $\mathbf{d}$, $\mathbf{x}$, and $\mathbf{E}$ reduced to their colinear components along D-H bond (see Methods, 'Full form of the dipole-in-$E$-field approximation'). Using Hooke's law, $d(E) = d(0) + F(E)/k(E)$, we can express both the electrostatic force $F(E)$ and the force constant $k(E)$ as derivatives of the potential energy around equilibrium as $F(E) = -\frac{\partial U_{HB}}{\partial d}$ and $k(E) = -\frac{\partial^2 U_{osc}}{\partial d^2}$, leading, in a linear approximation, to the $E$-field dependent equilibrium bond length $d(E)$ and force constant $k(E)$[17]:

$$d(E) = d(0) + \frac{E}{k(E)}\frac{\partial p}{\partial d} \qquad (3)$$

$$k(E) = k(0) - \frac{\partial^2 p}{\partial d^2}E \qquad (4)$$

Here, $d(0)$ and $k(0)$ are the equilibrium bond length and the force constant of a free D-H bond in the absence of the HB, respectively. In the harmonic approximation, the force constant $k(E)$ of D-H bond can be determined experimentally (using vibrational spectroscopy) from the stretching vibration frequency $\omega_{D-H}$ of the D-H bond, $k(E) = \mu\omega_{D-H}^2$, where $\mu$ is the reduced mass of the D-H system. Equations 3 and 4 give the first and second derivatives of the D-H dipole moments with respect to distance, enabling us to calculate the dipole moment of the hydrogen-bonded D-H system from Taylor expansion, and to derive parameters related to hydrogen-bonding confinement such as HB energy and dielectric properties.

### Experimental realization of the model in gypsum

To obtain the coefficient $\partial^2 p/\partial d^2$ in the Eq. 4, we performed vibrational spectroscopy experiments on water HBs in a solid-state system where the rotational degree of freedom of water molecules is suppressed. We chose gypsum ($CaSO_4 \cdot 2H_2O$) as our model system. Gypsum is one of the most abundant minerals on Earth and consists of alternating water bilayers and ionic $CaSO_4$ sheets[18,19]. In gypsum, water molecules are confined and orientated between $CaSO_4$ walls via HBs, forming a two-dimensional (2D) water crystal that extends along the gypsum a–c plane (Fig. 1c). We selected gypsum for three main reasons. First, the low dimensional hydrogen bonded water in gypsum offers a favorable platform to study 2D confined water – a

hot topic by its nature, and simplifies relevant theoretical analysis. Second, the stretching vibrations of the two O-H dipoles in the gypsum water molecules are sufficiently decoupled. In gypsum, water molecules are arranged in four orientations with two distinct types of HBs, resulting in two groups of O-H dipoles in different crystallographic environments[19,20]: O-H$_A$, bonded by a stronger intralayer HB, and O-H$_B$, with a weaker interlayer HB nearly parallel to the b-axis (Fig. 1d). Unlike the liquid water where the O atoms in water serve as both hydrogen donor and acceptor within small water clusters[21], the distance between O atom in one water molecule and H atom in another exceeds 2.8 Å, much greater than the typical length of H...O in conventional HB, suggesting minimal interactions between water molecules. Therefore in gypsum, the O atoms in one water molecule do not serve as HB acceptor for another, indicating the absence of intermolecular HBs within the 2D water layer. The water molecules in gypsum are bonded only to the $CaSO_4$ molecular walls. This has been confirmed by our DFT simulation. Additionally, we neglected the interaction between $Ca^{2+}$ ions and O atoms in water due to the considerable distance separating them (-2.3 Å). Figure 1e shows the Raman spectrum of the $H_2O$ stretching vibrations in a bulk gypsum crystal, with two peaks centered at 3405 cm$^{-1}$ and 3490 cm$^{-1}$ attributed to the stretching vibrations of O-H$_A$ and O-H$_B$ bonds, respectively (see details in Methods, 'Decoupling of the stretching frequencies of O-H bonds in crystalline water of gypsum'). The polar plot in Fig. 1f shows the angle-resolved Raman intensity of O-H$_A$ and O-H$_B$ modes with respect to the a-c plane of gypsum. The angles of maximal intensity are $\phi_E = 18°$ for O-H$_A$ and $\phi_E = 90°$ for O-H$_B$, with a difference of 72°. In subsequent experiments, we used these polarizations to obtain maximum O-H stretching signal.

Third, gypsum can be easily exfoliated into few-layer sheets, which are stable in ambient conditions. This enables us to apply van der Waals technology[22] to construct gypsum heterostructures, which allows us to apply external electric field $\mathbf{E}_{ext}$ while simultaneously probing the vibrational response of confined water. To this end, we sandwiched gypsum crystals (50–150 nm thickness) between hexagonal boron nitride (hBN) and few-layer graphene (FLG) to form heterostructures (Fig. 2a, b), resembling a parallel plate capacitor. The top and bottom FLGs allow us to apply $E_{ext}$ as a perturbation to the local field $E_{HB}$ of the acceptor A (in the case of gypsum, A is the oxygen atom in sulfate group, see Fig. 1b). We then measured Raman spectra of these two O-H stretching vibrations with the $E_{ext}$ field applied along the out-of-plane b-axis.

### Electrically tunable stretching vibration modes

The color maps in Fig. 2c, d show the evolution of the Raman spectra for O-H$_A$ and O-H$_B$ stretching vibrations as a function of $E_{ext}$ at 80 K, respectively. Both maps illustrate the impact of $E_{ext}$ on the O-H stretching vibration frequency, with O-H$_A$ showing pronounced peak splitting and O-H$_B$ showing peak shifting. These observed changes are attributed to alterations of the total electric field ($\mathbf{E}_{tot} = \mathbf{E}_{HB} + \mathbf{E}_{ext}$) acting on different types of O-H bonds in gypsum. The inset of Fig. 2e demonstrates how $\mathbf{E}_{tot}$ differs for O-H dipoles that point towards opposite directions. We observed similar behavior for the O-H$_A$ stretching in Fourier transform infrared (FTIR) spectroscopy measurements at low temperature (Supplementary Fig. 1). The observed effects from $\mathbf{E}_{ext}$ are reversible within our measurement range of ±0.5 V nm$^{-1}$, as the spectra revert to their original state when $E_{ext}$ was removed (Supplementary Fig. 2). This indicates that the field-induced changes in the spectra are elastic effects rather than permanent structural changes to gypsum. We also ruled out the possibility of spectrum evolution due to electrostatic pressure effects, as the pressure induced by $\mathbf{E}_{ext}$ is estimated to be -0.01 GPa at $E_{ext} = 0.5$ V nm$^{-1}$. According to reported values[23], such pressure would cause shifts of only 0.01 and 0.2 cm$^{-1}$ for O-H$_A$ and O-H$_B$, respectively, which are below our detection limit.

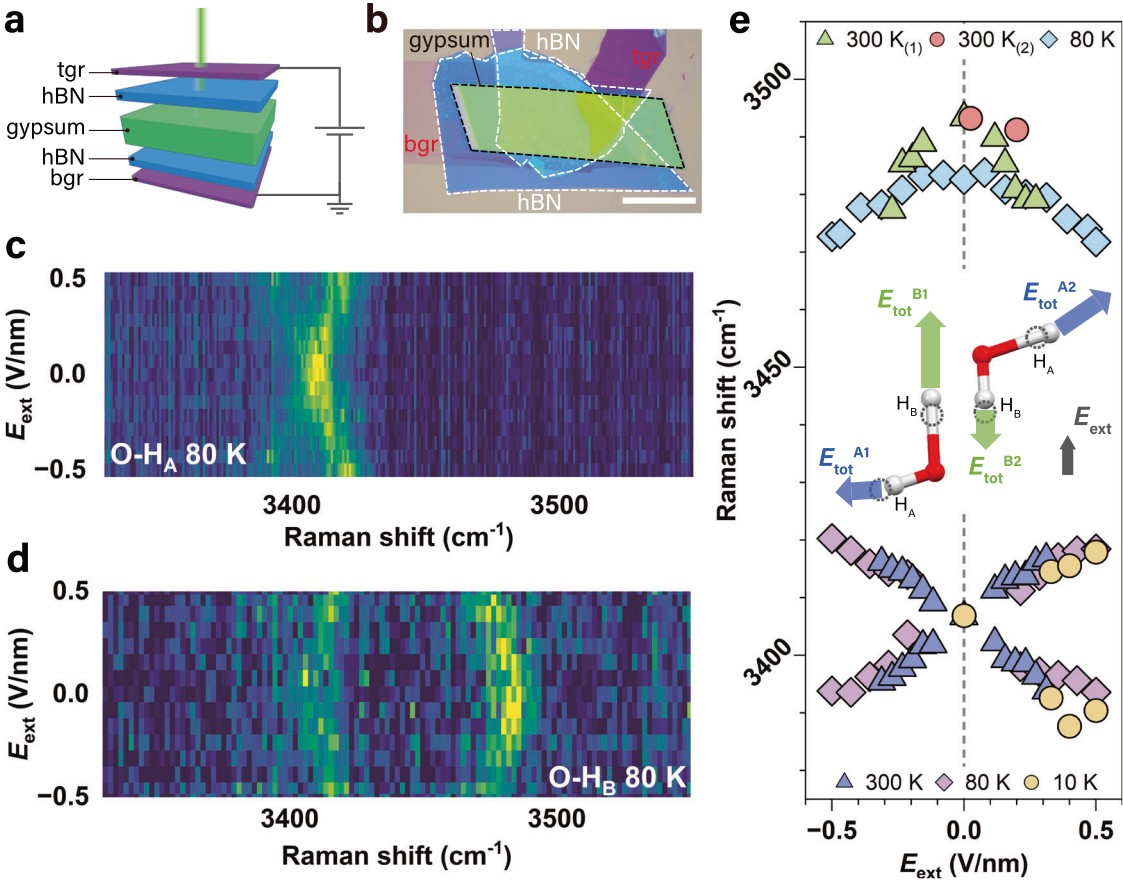

**Fig. 2 | Electric field effects on O-H dipole stretching vibration. a** Schematic of Raman measurements with external electric field, $E_{ext}$, applied to a tgr/hBN/gypsum/hBN/bgr heterostructure. **b** Optical micrograph of one of our gypsum heterostructure devices. tgr and bgr refer to top and bottom FLG electrodes. Scale bar is 20 μm. Color maps of Raman spectra for O-H$_A$ (**c**) and O-H$_B$ (**d**) at 80 K under $E_{ext}$. Color from dark blue to yellow indicates increasing Raman intensity. **e** Raman peak positions of the two O-H stretching modes at different temperatures under $E_{ext}$. Datasets with the same temperature but different colors were collected from different samples. Six datasets were acquired from six separate samples. The inset shows the change in O-H dipoles and $E_{tot}$ under $E_{ext}$, the effect is exaggerated for clarity. Dashed circles mark positions of H atoms in the absence of $E_{ext}$. Source data are provided as a Source Data file.

Figure 2e summarizes the dependence of the Raman peak positions of the O-H$_A$ and O-H$_B$ modes on $E_{ext}$. The procedures for extracting peak positions are detailed in Methods, 'Data analysis of Raman spectra.' O-H$_A$ stretching mode peak position is temperature insensitive, since the $E_{ext}$-induced splitting remained unchanged across the measured temperatures (10, 80, and 300 K). For O-H$_B$ stretching mode, however, a red-shift from 3497 to 3481 cm$^{-1}$ was observed at zero $E_{ext}$ when temperature decreased from 300 K to 80 K. The FTIR peak splitting of the O-H$_A$ stretching is summarized in Supplementary Fig. 1. The difference between Raman and IR peak positions at zero $E_{ext}$ arises from the frequency mismatch of the Raman-active mode ($A_g$) and IR-active mode ($A_u$), as confirmed by our DFT calculations (see Methods, 'Decoupling of the stretching frequencies of O-H bonds in crystalline water of gypsum' and Supplementary Fig. 3). We also investigated the H-O-H bending mode under $E_{ext}$ using scattering-type scanning near-field optical microscopy, as the bending mode provides direct access to the HB characteristics[24]. However, no $E_{ext}$-induced peak splitting or shifting was detected, consistent with our DFT results (see Methods, 'Water bending mode in gypsum' and Supplementary Fig. 4 for details).

**Fitting dipole-in-$E$-field model with experimental data**
Now, we recall the model at the beginning of the paper to rationalize our experimental results. First, we convert the $E_{ext}$-dependent stretching mode peak positions ($\omega_{D-H}$) into force constants, $k(E) = \mu\omega_{D-H}^2$, to obtain $k(E)$ as a function of $E_{ext}$. The total electric field acting on an O-H dipole (denoted as $E_{tot}$), is the sum of the internal $E$-field ($E_{HB}$) and the external electric field projected on the O-H dipole ($E_{ext}\cos\vartheta$):

$$E_{tot} = E_{HB} + E_{ext}\cos\theta \qquad (5)$$

Here, $\theta$ is the angle between and the gypsum b-axis (Fig. 1b). For the dipoles O-H$_A$ and O-H$_B$ in gypsum, $\theta$ is 69.60° and 9.37°, respectively, as obtained from neutron diffraction data[20]. For a free water molecule, $k(0) = 762.0 \pm 0.3$ N m$^{-1}$ [25]. From here, we construct a plot of $k(E)$ vs $E_{tot}$, Fig. 3a. Further details of data processing and error analysis are in Methods, 'Fitting of $k$ vs $E$ relation in Fig. 3a and local fields at dipoles O-H$_A$ and O-H$_B$.' This analysis allows us to quantify the internal electric field $E_{HB}$ for O-H$_A$ and O-H$_B$ as 5.33 ± 0.63 V nm$^{-1}$ and 3.82 ± 0.45 V nm$^{-1}$, respectively. From Eq. 4, we obtain the slope of $k(E)$ vs $E_{tot}$ curve in Fig. 3a, the second derivative of the dipole moment along the O-H bond, $\frac{\partial^2 p}{\partial d^2} = (2.2 \pm 0.3) \times 10^{-8}$ C m$^{-1}$.

Similarly, we further obtain the first derivative of the dipole moment, $\frac{\partial p}{\partial d}$, using Eq. 3. To do this, we rely on existing literature data of several solid-state systems containing crystalline water, such as proton-ordered and proton-disordered ices, and solid hydrates. For free water molecule, we used the equilibrium O-H bond length obtained from fitting databases of experimental rotational energy levels[26,27]. For neutron diffraction data we applied thermal corrections, more details are in Methods, 'Thermal corrections for neutron structural data.' From here, we establish the $d(E)$ vs $E/k(E)$ relation, Fig. 3b,

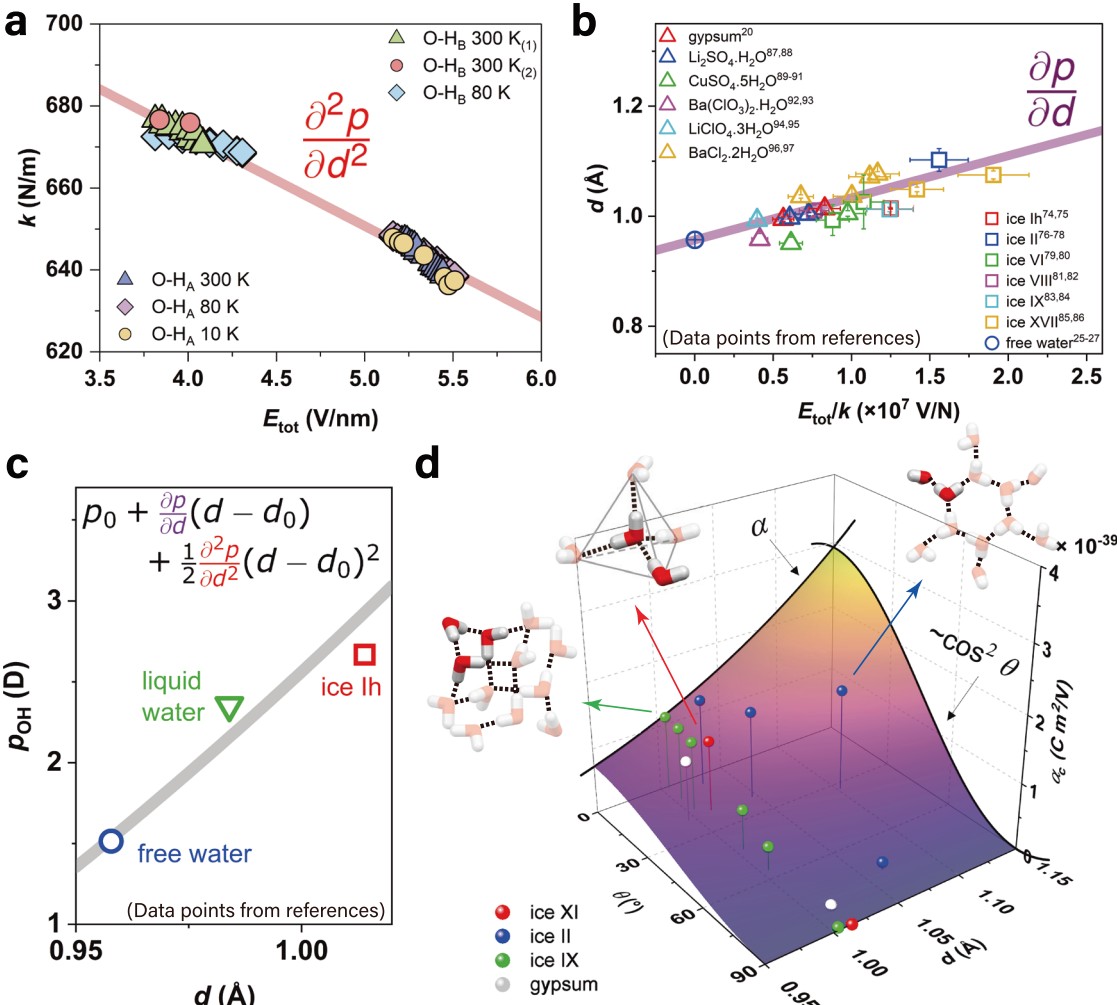

**Fig. 3 | Hydrogen bonding and dielectric properties of water. a** Force constant $k$ vs $E_{tot}$ along O-H bond. Symbols represent experimental results from this work using gypsum at different temperatures. The linear fit using Eq. 4 (red line) gives the second derivative of the dipole moment along the O-H bond, $\partial^2 p/\partial d^2$. **b** O-H bond length $d$ versus $E_{tot}/k$ for a range of solid-state water systems from literature (references are in Methods, 'Data analysis for bond length vs $E/k$ relation in Fig. 3b'). Error bars for $d$ were provided in the references; error bars for $E_{tot}/k$ were obtained from the calculations using our fitted $\partial^2 p/\partial d^2$ from (**a**). Linear fit using Eq. 3 (purple line) gives the first derivative of the dipole moment, $\partial p/\partial d$. **c** Estimated dipole moment along O-H bond, $p_{OH}(d)$, constructed using Eq. 6 (gray line), plotted

alongside with different water systems from the literature (for details and references see Methods, 'Dipole moments vs O-H bond lengths in Fig. 3c'). **d** Effective polarizability along the confining direction ($\alpha_c$) of a single confined O-H dipole against the O-H bond length $d$ and the angle $\vartheta$ between the dipole and the confining direction. $\alpha$ is the polarizability of O-H along dipole direction ($\vartheta = 0°$) given by Eq. 7. Red, blue, green, and gray dots represent the O-H dipoles in proton-ordered ice XI, II, IX, and gypsum. The arrows point to the structures of ice XI, II, and IX, with the geometrically non-degenerate water molecules highlighted. Source data are provided as a Source Data file.

using systems for which literatures contain both neutron diffraction data to obtain O-H bond length $d$, and Raman spectra to get $k_E$ and $E$ from O-H stretching vibration frequency. Linear fitting of Fig. 3b gives the first derivative of dipole moment, $\frac{\partial p}{\partial d} = (7.7 \pm 1.2) \times 10^{-19}$ C (Eq. 3), and the fitted O-H bond length of a free water molecule as $d(0) = 0.9570$ Å (literature sources, details of data analysis, as well as fitting details are in Methods, 'Data analysis for bond length vs $E/k$ relation in Fig. 3b'). Given that both data from solid hydrates and different phases of ices have been used in the fitting in Fig. 3b, our model is designed to be applicable to both solid hydrate and ice systems.

We then reconstruct a universal expression for O-H dipole moment $p(d)$ using the obtained derivatives in a Taylor expansion:

$$p(d) = \sum_{n=0}^{\infty} \frac{1}{n!} \frac{\partial^n p}{\partial d^n} (d - d(0))^n \qquad (6)$$

By using the first two derivatives ($\frac{\partial^2 p}{\partial d^2}$ and $\frac{\partial p}{\partial d}$) obtained from Fig. 3a, b, and the known O-H dipole moment for a free water molecule

$p(d(0)) = 1.5140 \pm 0.0004$ D[26–31], we plot the O-H dipole moment $p_{O-H}$ as a function of O-H bond length in Fig. 3c. The calculated dipole moment $p_{H2O} = 1.88 \pm 0.04$ D for free water molecule agrees well with the reported value $p_{H2O} = 1.855 \pm 0.001$ D[29]. For liquid water, our model predicts $p_{H2O} = 2.66 \pm 0.13$ D, aligning well with the reported values of $2.9 \pm 0.6$ D[32] within uncertainty range. For ice Ih, our model estimated a dipole moment of $3.39 \pm 0.24$ D for a proton-order ice Ih structure, which is higher than the dipole moment reported for proton-disordered ice Ih, $3.09 \pm 0.03$ D, derived from the multipole moments expansion of water molecule[33]. The overestimated dipole moment may result from the absence of defects that are typically present in ice Ih. These defects facilitate the formation of $H_3O^+$ and $OH^-$ pairs and enable proton hopping in ice Ih, which can weaken the HBs[34]. To exclude the effect from defects, we compared our results with those from the proton-ordered counterpart of ice Ih, commonly known as ice XI. The dipole moment of $3.39 \pm 0.24$ D estimated by our model aligns closely with the 3.3 D reported for water molecules in ice XI, given by DFT simulations[35]. All estimated dipole moments together

with reference values are summarized in Supplementary Table 2. For details about dipole moment calculations, see Methods, 'Dipole moments vs O-H bond lengths in Fig. 3c.'

## Dipole-in-$E$-field model for hydrogen bond energy

Further, we estimate the HB energy $U_{HB}$ (Eq. 2) using the derived dipole moment and electric field values, given by Eqs. 6 and 4, respectively. For example, Ice Ih has a Raman shift of 3279 cm⁻¹ and a bond length of 1.014 Å (see Methods, 'Data analysis for bond length vs $E/k$ relation in Fig. 3b'). Liquid water has a bond length of 0.984 Å (see Methods, 'Dipole moments vs O-H bond lengths in Fig. 3c') and a broad O-H stretching band due to the extensive HB network, ranging from 3200 cm⁻¹ to 3650 cm⁻¹ [36,37]. Using our approach, we obtain $U_{HB}$ per HB as 10.5 ± 1.2 kcal mol⁻¹ (455 ± 52 meV) for ice Ih and 5.1 ± 0.5 kcal mol⁻¹ (221 ± 22 meV) for liquid water. Within the uncertainty range, our $U_{HB}$ for liquid water is consistent with 5.26 kcal mol⁻¹, the reference value of the liquid water HB strength estimated from the enthalpy of vaporization of water at 25 °C [38]. However, our estimation for ice Ih is higher than the reported lattice energy, 7.04 kcal mol⁻¹ (305 meV) [39,40]. This difference could be due to an overestimated dipole moment calculated on a proton-ordered ice Ih structure where H atoms have fixed positions, as discussed above. In the real ice Ih, protons are disordered and can hop between different crystalline sites, which tends to weaken the HB strength. Such weakening effect is reflected in our model by the overestimated $U_{HB}$ for ice Ih. All estimated $U_{HB}$ and the comparison with reported values are summarized in Supplementary Table 2.

## Predicting the dielectric properties of HB systems

Beyond HB characterization such as dipole moment and HB energy, our approach could also predict other properties of the HB system, such as polarizability and dielectric constant. Below we show briefly the prediction power of our approach. For example, polarizability along the O-H dipole, $\alpha(d) = \partial p/\partial E$, can be evaluated using the relationship $d(E)$ from Eq. 3, as,

$$\alpha(d) = \left[\frac{\partial p}{\partial d} + \frac{\partial^2 p}{\partial d^2}(d - d(0))\right]^3 / \frac{\partial p}{\partial d} k(0) \quad (7)$$

Here, $\frac{\partial p}{\partial d}$ and $\frac{\partial^2 p}{\partial d^2}$ derived from our model, and $d(0)$ and $k(0)$ taken from the free water molecule (see Methods, 'Fitting of $k$ vs $E$ relation in Fig. 3a and local fields at dipole O-H$_A$ and O-H$_B$'). In nanoconfined and crystalline water systems, the rotational degrees of freedom of the dipole are restricted, meaning that no dipole reorientation under $E_{ext}$ will contribute to the dielectric response. Therefore, Eq. 7 fully describes the polarizability of such systems. The effective polarizability along b-axis is given by $\alpha_c = \alpha \cos^2\theta$ and is determined by both $d$ and $\vartheta$, as summarized in Fig. 3d, along with exemplar nanoconfined water systems. For system with known bonding geometry and bond length $d$, the polarizability of each O-H dipole along any confining axis can be evaluated via Fig. 3d. And the polarizability along the confining direction can be obtained by summing up $\alpha_c$ for all dipoles in the system.

From here, we further extract the static dielectric constant $\varepsilon_r$ of a system consisting of electrostatically confined water molecules (see also Methods, 'Polarizability and dielectric behavior of confined water'). In such a system, the polarizabilities of individual O-H dipoles add up to give $\varepsilon_r$:

$$\varepsilon_r = 1 + \frac{1}{\varepsilon_0}\sum_i \alpha_{ci}N_i \quad (8)$$

where $\alpha_{ci}$ is the effective polarizability along confining direction and $N_i$ is the number density of the $i^{th}$ type of O-H dipoles. Using this relation, we obtain $\varepsilon_r = 6.6 \pm 0.7$ for the confined 2D water layer in gypsum. The

low $\varepsilon_r$ of the 2D water sheets is consistent with the decreasing $\varepsilon_r$ of confined liquid water along perpendicular direction, due to restricted rotational freedom [41,42]. Our capacitance measurements of gypsum crystal give $\varepsilon_r = 3.68$ (see Methods, 'Polarizability and dielectric behavior of confined water' and Supplementary Fig. 5). Since gypsum is an hydrogen bond heterostructure (HBH) composed of alternating $CaSO_4$ and water dielectrics, we can evaluate $\varepsilon_r = 2.8 \pm 0.5$ for the bare $CaSO_4$ molecular sheets, which matches well with the reported $\varepsilon_r$ of β-anhydrite, 2.7 ± 0.02 [43]. The estimated $\varepsilon_r$ for gypsum system is summarized in Supplementary Table 2.

Our model also accurately replicates the dielectric constants for three proton-ordered ices: ice XI (ordered ice Ih), ice II, and ice IX (ordered ice III). The polarizability of O-H bonds is given by $\alpha_c = \alpha \cos^2\theta$ and Eq. 7, as shown in Fig. 3d (see also Methods, 'Dielectric constants of water ice systems'). For bulk liquid water, using the dipole moment from Eq. 6 and considering rotational averaging, we obtain $\varepsilon_r = 27 \pm 2$, which is much lower than experimental value of 79 (see Methods, 'Dielectric constant of liquid water'). This discrepancy arises because the rotations of water molecules are not independent of each other, an effect usually accounted for by introducing the dipole correlation factor $G_k$ [44]. To reproduce the experimentally obtained $\varepsilon_r$, $G_k \approx 3.15$ is needed in our case, falling in the range of calculated value of 2.72–3.70 for bulk liquid water [45]. Our model also cannot predict the anomalously high dielectric constant in proton-disordered ices, such as $\varepsilon_r$ on the order of 100 for ice Ih, the most common form of ice in nature [46,47]. This behavior is usually attributed to the presence of defects in the form of partial rotations and proton jumps, which form polarized domains at lower temperatures where ice sometimes becomes ferroelectric [48].

Our approach also reproduces the anomalously low $\varepsilon_{r,\perp}$ of confined water, which arises from the restricted rotational degrees of freedom of liquid water under nanoconfinement [49]. Here, we consider the dielectric response of confined water resulting only from electronic and atomic polarization, as described by Eq. 8. The dependence of $\varepsilon_r$ along the confining direction on the orientation of water molecule is plotted in Supplementary Fig. 6, calculated using the density of confined water 3.39 g cm⁻³ from recent simulations [50]. We found that when the water molecular plane is nearly parallel to the confining plane (confining angle $\theta < 23°$), we can reproduce the experimentally obtained $\varepsilon_{r,\perp} = 2.1$ [49].

## Beyond conventional HB systems

Our dipole-in-$E$-field approximation can be generalized to other types of electrostatic interactions beyond conventional HBs. Instead of conventional chemical bonds, our model conceptualizes HBs as universal electrostatic interaction between an O-H dipole and electric field from acceptor. Consequently, although the experimental data was collected on solid hydrate like gypsum, our approach can be generalized to any systems based on electrostatics interactions containing O-H dipole, including conventional hydrogen bonding systems like liquid water and ice, and π-hydrogen bonding systems such as water-carbon interface. Once the vibrational frequency of water under confinement is known, the strength of the electrostatic interaction between water and its surroundings can be evaluated. For example, interfaces between water and carbon materials often exhibit unexpected phenomena such as low friction or long slip length, which are promising for water-energy applications [51]. Such electrostatic interactions between water O-H bonds and the π orbitals of sp²-hybridized carbon can be viewed as weak HBs, evidenced by red-shifts in the water stretching vibration frequency. Using our approach, we can evaluate the interaction energy between water O-H dipoles and π orbitals at graphitic surfaces.

For instance, in the case of single water molecules trapped inside a $C_{60}$ fullerene [52], from the measured symmetric $\upsilon_1$ and antisymmetric $\upsilon_3$ stretching modes at 3573 cm⁻¹ and 3659.6 cm⁻¹, we can calculate interaction strength $U_{HB} = 1.5 \pm 0.2$ kcal mol⁻¹ per bond (66 ± 9 meV),

comparable to typical weak HBs. Similar values are obtained for water confined in single-walled carbon nanotubes[53]. Further evidence supporting our approach comes from recent sum frequency generation (SFG) measurements at the water-graphene interface[54], where the authors reported vibration frequency of $\nu_{OH} = 3616$ cm$^{-1}$ for the dangling O-H dipole. Our approach predicts the HB energy in this case very close to that for water confined in C$_{60}$. More conclusively, the stretching frequency of this dangling O-H bond shows an electric field tuning effect that matches perfectly with our model (see Methods, 'Water-carbon interface.' and Supplementary Fig. 7). More details on evaluating water-carbon interactions see Methods, 'Water-carbon interface.'

## Discussion

Our dipole-in-$E$-field approach reliably quantifies the characteristics of confined water molecules in various complex environments, advancing our understanding of HB in biologically and technologically relevant systems. Using coefficients parametrized from this work, we can quantitatively evaluate properties such as O-H bond length, local electric field, dipole moment, HB strength, polarizability, and dielectric constant directly from spectroscopic data (Raman, FTIR, or SFG) for a wide range of water systems, including crystalhydrates, ices, and nanoconfined water. This approach can be extended beyond confined water to systems such as organic molecules, (bio)polymers, liquid phases, and electrochemical interfaces, which are crucial for bio-medicine and technology. Testing the limits of our approach in scenarios where quantum and many-body effects dominate, such as blue-shifting strong HBs[7] or cooperative HB systems[55], could provide new insights into the fundamental nature of hydrogen bonding.

Notably, our work represents the first study of hydrogen bond heterostructures (HBHs). While previous research has explored heterostructures composed of hydrogen-bonded organic frameworks (HOFs) and graphene, highlighting how intralayer HBs within the HOF enhance interactions with graphene[56], our study primarily focuses on the interlayer HB that binds layers along out-of-plane direction, which could emerge as a class of intelligent materials allowing for facile assembly and fabrication, similar to van der Waals materials. The highly anisotropic HBHs—which potentially can be probed using dichroic IR measurement[57]—offer unique properties, including electrical and chemical tunability, stronger and more directional bonding compared to van der Waals forces, and the potential for self-assembly and emergent phenomena. By complementing hydrogen-bonded organic frameworks, HBHs and HB technology could enhance applications in gas separation, catalysis, and (opto)electronic devices[58-63]. The combination of our dipole-in-$E$-field approach and the development of HBHs opens new avenues for the rational design and precise control of the properties of hydrogen-bonded systems.

## Methods

### Full form of the dipole-in-$E$-field approximation

The 3D form of Hooke's law can be written as $\mathbf{F} = -\boldsymbol{\kappa} \cdot \mathbf{X}$. Here $\mathbf{X}$ is the displacement vector in 3D space, and $\boldsymbol{\kappa}$ is a $3 \times 3$ strain stiffness tensor, where the diagonal and off-diagonal indices are the stiffness in directions parallel and normal to each force components. The potential energy of a harmonic oscillator in 3D case is:

$$U_{osc} = \frac{1}{2}\mathbf{X}^{\mathbf{T}} \cdot \boldsymbol{\kappa} \cdot \mathbf{X} + U_E, \qquad (9)$$

$$U_{HB} = -\mathbf{p} \cdot \mathbf{E} + \dots \qquad (10)$$

Set x-axis along D-H bond, then we have electrostatic force and sum potential stiffness: $\mathbf{F}(\mathbf{E}) = -\boldsymbol{\nabla}U_{HB}$ and $k(\mathbf{E}) = \nabla^2 U_{osc}$. The

field-dependent bond length and field-dependent spring constant are then:

$$\mathbf{d}(\mathbf{E}) = \mathbf{d}(0) + \nabla \mathbf{p} \cdot \mathbf{E} \frac{1}{k(\mathbf{E})} \qquad (11)$$

$$k(\mathbf{E}) = k(0) - \nabla^2 \mathbf{p} \cdot \mathbf{E} \qquad (12)$$

In this work, we consider only the parallel response along D-H bond. Thus, among the nine indices of $\boldsymbol{\kappa}$, only one diagonal term is non-zero. The system then simplifies to the 1D problem described in the main text.

The electric field in our model is defined as the effective electric field at the dipole position, generated by the hydrogen acceptor, A, with the modification from dipole-field interactions inherently accounted for. As the dipole moves within acceptor's field, the interaction induces a charge redistribution in the acceptor, dynamically evolving the field and continuously affecting the HB strength. However, these effects do not require separate evaluation. Using the dipole-in-field model, the HB energy is directly determined as the product of the effective dipole moment and the effective electric field. Consequently, the modification of $\mathbf{E}$ due to dipole-field interactions is incorporated into our model by treating $\mathbf{E}$ in Eq. 2 as the effective electric field at the dipole position, where the contribution of $\mathbf{p}$ has been embedded. This effective $\mathbf{E}$ also corresponds to the Raman peak shift measured experimentally.

### Device fabrication

Gypsum films were isolated from bulk natural crystals onto SiO$_2$ (290 nm)/Si and CaF$_2$ substrates (for spectroscopic measurements) and polydimethylsiloxane (PDMS) stamps (for heterostructure fabrication) using micromechanical exfoliation in cleanroom. Graphite/gypsum/graphite heterostructure is fabricated using all-dry viscoelastic stamping technique[64]: first, graphite flake (bottom electrode) was isolated from natural graphite crystal onto SiO$_2$/Si substrate; gypsum flake and top graphite electrode were exfoliated onto PDMS stamp. The gypsum and top graphite flakes were then transferred onto bottom graphite electrode successively, aligning and stacking target flakes under the optical microscope using transfer rig.

Gypsum is prone to dehydration in vacuum or when exposed to elevated temperature, therefore, conventional lithography and metal deposition process cannot be used to fabricate electrical contacts to gypsum. Instead, we used two thick graphite flakes with over 100 μm length as extended electrical contacts to the top and bottom FLGs of the heterostructure, to which gold wires were then attached using conductive silver paint.

For devices with hBN separation layers, the hBN flakes were exfoliated and transferred using polypropylene carbonate (PPC) dry transfer technique[65]: the hBN flakes exfoliated on SiO$_2$/Si wafer were picked up using PPC coated PDMS stamp at 45 °C, and then aligned and transferred onto desired stacks at 60 °C under the optical microscope using transfer rig. All fabrication processes were carried out in a cleanroom.

### Vibrational spectroscopy measurements

Angular-resolved polarized Raman spectroscopy was performed using Oxford WITec alpha300R confocal Raman system, with 600 lines mm$^{-1}$ grating and a 532 nm wavelength laser with linear polarization. Laser power was kept below 8 mW. During measurements, the sample was fixed on the sample stage, with incident laser polarization rotated by a polarizer at a step of 5°. The spectra were taken with parallel analyzer configuration at each incident orientation, so the collected light had the same polarization direction as incident. For each spectrum, the acquisition time was 10 s with 6 accumulations.

Raman measurements under electric field in ambient conditions were carried out using Renishaw inVia confocal Raman system equipped with 2400 lines mm$^{-1}$ grating and 514 nm linearly polarized laser with power restricted below 8 mW, focused through a ×100/0.90 NA objective. The voltage was applied using a Keithley 2614B source meter.

Peak splitting is not very pronounced even at 0.3 V nm$^{-1}$ (e.g., Supplementary Fig. 8 for O-H$_A$ stretching mode), but higher field is unreachable at ambient conditions due to field-induced degradation of FLG electrodes, even with hBN spacers. So, we performed Raman measurement at T < 80 K, where $E_{ext}$ of up to 0.5 V nm$^{-1}$ can be applied and distinct peak splitting is well resolved.

Raman measurements at cryogenic temperatures were performed on WITec alpha300 R Raman system with 600 lines mm$^{-1}$ grating and ×100/0.85 NA objective using 532 nm linearly polarized laser with power lower than 8 mW. The sample was placed in an Oxford microstat system in vacuum with temperature controlled by a liquid N$_2$ circulation loop for measurements at 80 K and liquid He for measurements at 10 K. The fluctuation of temperature was ±5 K. Due to the light absorption by the optical window on Oxford microstat system, the acquisition time was extended to between 30 and 60 min, depending on the signal to noise ratio on different devices, to increase the Raman intensity. This significantly increased the measurement duration, so no spatial mapping data was collected while the external electric fields were applied. To protect the samples from potential laser-induced damage, the laser power for all Raman measurements was maintained below 8 mW.

Fourier transform infrared (FTIR) spectra were recorded with a Bruker VERTEX 80 spectrometer equipped with a Bruker HYPERION 3000 FTIR microscope. For low temperature FTIR measurements at 83 K, the sample was placed in a Linkam stage with temperature controlled by a liquid N$_2$ circulation loop. All spectra were collected with 4 cm$^{-1}$ resolution and averaged over 512 scans.

Nano-FTIR measurements were performed on an AFM based scattering-type scanning near-field optical microscopy (s-SNOM) from neaspec GmbH. The conventional Pt/Ir probe (ARROW-NCPt-50, Nanoworld) with a tapping frequency of 270 kHz was illuminated with a p-polarized mid-infrared broadband difference frequency generation (DFG) laser (frequency range 1000–1900 cm$^{-1}$, average power <1 mW). The sample for s-SNOM measurements was FLG/gypsum/FLG stack on SiO$_2$/Si substrate. During measurements, a dc voltage was applied to bottom graphite electrode, while the top graphite and AFM probe were electrically grounded, creating an electric field along out-of-plane direction. Obtained nano-FTIR spectra were collected at different voltage levels by averaging 30 interferograms (spectral resolution ≈7 cm$^{-1}$) with 2048 pixels and an integration time of 10 ms/pixel. All spectra are normalized to that of a Si substrate.

## Data analysis of Raman spectra

For Fig. 1f in the main text, we first performed background subtraction on raw spectra using 2nd polynomial with 256 sample points, followed by peak fitting using Voigt function. The peak position and peak width of water stretching modes were constrained within 3400–3412 cm$^{-1}$ and 20–50 cm$^{-1}$ for O-H$_A$, and 3490– 3500 cm$^{-1}$ and 20–70 cm$^{-1}$ for O-H$_B$. Raman spectra in Fig. 2 (main text) were obtained first through baseline fitting and subtraction using 4th polynomial with 256 sample points, followed by peak fitting using Voigt function. Initial peak positions were set at the position with maximum intensity. The peak widths were limited within 100 cm$^{-1}$ and 90 cm$^{-1}$ respectively.

## Decoupling of the stretching frequencies of O-H bonds in crystalline water of gypsum

In gypsum, water O-H covalent bonds are stretched due to HB. Different crystalline environment of the two O-H bonds results in a distorted asymmetric water molecule. Such asymmetry decouples and

localizes the normal symmetric and antisymmetric stretching onto individual O-H bonds. Consequently, one O-H bond will have much larger vibration amplitude than the other in each vibration mode.

We performed density functional theory (DFT) calculations using Quantum ESPRESSO package, to elaborate on the vibrational response of gypsum. Dispersion corrections that account for the van der Waals forces were incorporated following the empirical D3 formulation by Grimme, yield 0.06% difference with experimental lattice parameters. Calculations of phonon frequencies were carried out using the finite difference methodology implemented in Phonopy by displacing each atom 0.001 Å along three principal axes. Generalized Gradient Approximation (GGA) formulation by Perdew, Burke and Ernzerhof was used to model exchange-correlation interactions among electrons in all the calculations. An energy cutoff of 65 Ry with a uniform Monkhorst-pack k-mesh of 5 × 5 × 5 is used for all the calculations on the primitive unit cell. Convergence criteria used for all self-consistent field calculations are 10$^{-10}$ Ry per formula unit.

In gypsum crystal, water has 8 stretching modes in total, Supplementary Fig. 3. According to our DFT calculations, they are grouped into two branches, high- and low-frequency (near 3500 cm$^{-1}$ and 3400 cm$^{-1}$), with vibration amplitudes mainly localized on O-H$_B$ and O-H$_A$ bonds, respectively. In each of these two branches, the four levels arise due to the symmetry of gypsum crystal. In particular, four water molecules in a unit cell are grouped into two inversion symmetric pairs. Each of the two water molecules in the pair vibrate collectively either in-phase or out-of-phase, which decouples the modes. In turn, each in-phase and out-of-phase modes has symmetric and antisymmetric intramolecular solutions, which leads to further mode decoupling.

## Infrared spectroscopy of water O-H$_A$ stretching modes in gypsum

Similar to Raman data presented in Fig. 2 in main text, peak broadening and splitting were also observed in low temperature FTIR measurements (Supplementary Fig. 1). The IR peak of O-H$_A$ mode blue-shifted from 3401 cm$^{-1}$ to 3412 cm$^{-1}$ when temperature reduced from 300 to 80 K, possibly due to a volumetric contraction effect which shortens the effective O-H bond length[66]. Similarly to Raman, the O-H$_A$ stretching splits into two peaks, corresponds to the two O-H$_A$ $A$ modes in Supplementary Fig. 1b ($A_g^A$ and $A_u^A$ at zero field, Supplementary Fig. 3). The obtained electric field dependence of stretching peak position in FTIR spectra is similar to that of our Raman results.

Due to the large overlap between the two $E_{ext}$ induced O-H$_A$ peaks, we instead chose positions of maximal peak intensity for spectra fitting. At high $E$-fields, both O-H$_A$ peaks show similar splitting with electric field. However, at low $E$ fields, the low-frequency peak is nearly $E$-field independent. According to our DFT calculations, the two O-H$_A$ peaks correspond to two $A$ modes of O-H$_A$ at 3406 and 3393 cm$^{-1}$, which are only Raman-active ($A_g^A$ mode) and IR-active ($A_u^A$ mode) at zero field, respectively. They are not the same mode and have different frequency. This accounts for the absence of $A_g^A$ mode in IR and $A_u^A$ mode in Raman at zero field, and the deviation of IR data from linear splitting at low $E_{ext}$-field (<0.065 V nm$^{-1}$) in Supplementary Fig. 1c.

## Water bending mode in gypsum

Due to extremely low Raman intensity, water bending mode and its field-dependence were investigated via nano-FTIR measurements using scattering-type scanning near-field optical microscopy (sSNOM). Water in gypsum exhibits two bending modes located around 1620 and 1680 cm$^{-1}$, Supplementary Fig. 4. This agrees well with our DFT calculations. The water bending mode also decouples into four sub-modes, because of four water molecules in a gypsum unit cell. Similar to the stretching vibration mode decoupling, inversion symmetry divides these four modes into two pairs, with frequencies around 1620 and 1680 cm$^{-1}$. We did not observe any $E$-field response for the bending

modes, Supplementary Fig. 4. This is consistent with our DFT calculations, which show that bending mode frequencies do not change even at a high electric field up to 1.8 V nm$^{-1}$, because phonon dispersion around Γ point was unaffected by external electric field.

## Fitting of k vs E relation in Fig. 3a and local fields at dipole O-H$_A$ and O-H$_B$

The $k$ vs $E$ relation in Fig. 3a in the main text was obtained by accounting for the applied $E_{ext}$ component with respect to O-H dipoles and accommodating the crystal structure symmetry of gypsum, using data from Fig. 2e, main text. To this end, the $E_{ext}$ for each dataset was scaled by cos $\vartheta$ to obtain the electric field component along each O-H dipole. Here $\vartheta$ (as defined in Fig. 1b, main text) is 69.60° and 9.37° for O-H$_A$ and O-H$_B$ dipoles, respectively[20]. Then, half of each dataset (those with positive slopes) were flipped horizontally, so that all the datasets having negative slope (red-shift). According to Eq. 4 in main text, this slope corresponds to $-\frac{\partial^2 p}{\partial d^2}$, where $E$ is the total electric field at the dipole, equal to $E_{HB} + E_{ext}$. All datasets transformed by the above procedure were then fitted globally using Eq. 13 with three fitting parameters—one is the shared slope $\frac{\partial^2 p}{\partial d^2}$ for both O-H$_A$ and O-H$_B$ data because $\frac{\partial^2 p}{\partial d^2}$ is expected to be the same for both dipoles, and two different $E_{HB}$ for O-H$_A$ and O-H$_B$ due to the different local HB field:

$$k(E) = k(0) - \frac{\partial^2 p}{\partial d^2}(E_{HB}^{A(B)} + E_{ext}\cos\theta) \qquad (13)$$

To obtain the intercept $k(0)$ in Eq. 4 in main text (and Eq. 13), we resorted to the known data on water in gas phase from literature. Vibrational spectroscopy of free water molecules is complicated: three vibrational modes—symmetric $v_1$ and antisymmetric $v_3$ stretching and bending mode $v_2$ together with numerous rotational modes result in hundreds of thousands of peaks of various intensities and linewidths[67–70]. The most recent values of free H$_2^{16}$O molecule's vibrational transition frequencies from the W2020 database are 3657.053, 3755.929, and 1594.746 cm$^{-1}$, for $v_1$, $v_3$, and $v_2$, respectively[25]. The vibration energy of uncoupled O−H bond $v_{O-H}$, estimated as the average of $v_1$ and $v_3$, is 3706.491 cm$^{-1}$, in good agreement with $v_{O-H}$ = 3707.467 cm$^{-1}$ for HD$^{16}$O molecule, where decoupling is provided by a large mismatch in effective masses of H and D[71]. This gives us the harmonic spring constant $k(0) \approx 762.0 \pm 0.3$ N m$^{-1}$ for uncoupled O−H bond in HB-free water. Linear fitting using Eq. 13 gives slope $-\frac{\partial^2 p}{\partial d^2} = (2.23 \pm 0.26) \times 10^{-8}$ C m$^{-1}$. The local $E_{HB}$-fields at O-H$_A$ and O-H$_B$ dipoles are $(5.33 \pm 0.63)$ and $(3.82 \pm 0.45)$ V nm$^{-1}$, respectively. The uncertainty is given by the 2$\sigma$ value of the linear fit (reduced $\chi^2$ = 2.3).

## Thermal corrections for neutron structural data

Bond lengths from neutron diffraction are based on time-averaged neutron scattering intensity, giving an averaged interatomic separation. The actual bond length is slightly longer, after applying thermal corrections. The thermally corrected O-H bond lengths in Fig. 3b in the main text were obtained in different ways. For ice Ih, ice VIII, ice IX, Li$_2$SO$_4$·H$_2$O, Ba(ClO$_3$)$_2$·H$_2$O, and LiClO$_4$·3H$_2$O, their thermally corrected O-H bond lengths were provided in the respective references. For ice II, ice VI, ice VIII and ice XVII, the thermal corrections were evaluated using isotropic temperature factor, $B$. For CuSO$_4$·5H$_2$O and BaCl$_2$·2H$_2$O, thermal corrections were carried out using anisotropic temperature factors, $\beta_{ij}$. The temperature parameters of the systems are given in corresponding references respectively. The approaches used to evaluated the thermal corrections were reported in ref. 72.

## Data analysis for bond length vs E/k relation in Fig. 3b

The various systems in Fig. 3b, main text contain ices and solid hydrates. The distance between H and metal ions (M) in most solid hydrates ranges from 1.5 to 3 Å. Here, we focused on pure hydrogen bonding interactions and analyzed the solid hydrates systems with H...M distance of 2–3 Å, ignoring materials such as BeSO$_4$·4H$_2$O, where the significantly shorter H...Be distance of 1.6 Å may affect the charge distribution on H via electrostatic interactions from Be cation. All O-H bond lengths are from thermally corrected neutron diffraction data. Where the O-H length was measured on deuterated water D$_2$O, the thermal-corrected bond length is further upshifted by 3% to account for the bond difference between H$_2$O and D$_2$O[73]. For vibrational spectroscopy data on systems with multiple stretching peaks, we assigned peaks to O-H bonds with different bond lengths such that a shorter bond has a higher vibration frequency due to the steeper potential profile. The assignment process for each confined-water system is discussed below and data is summarized in Supplementary Table 1.

For ice Ih, one Raman peak is observed at 3279 cm$^{-1}$ for 3 mol% HOD ice at 123 K, corresponds to the stretching vibration of O-H bond, free of intra-molecule coupling due to large differences in the atomic mass of H and D[74]. Two O-H bond length of $(1.008 \pm 0.002)$ Å and $(1.004 \pm 0.001)$ Å given by neutron diffraction at 123 K were thermally corrected by +0.008 Å and averaged to $(1.014 \pm 0.0011)$ Å[75].

For ice II, two strong Raman peaks were found at 3194 and 3314 cm$^{-1}$ at 77 K[76,77]. Neutron diffraction of D$_2$O ice II gives four O-D bond lengths of $(1.014 \pm 0.020)$ Å, $(0.975 \pm 0.020)$ Å, $(0.956 \pm 0.020)$ and $(0.937 \pm 0.020)$ Å on two symmetrically non-degenerated water molecules at 110 K, with isotropic temperature parameter ($B$) of 4.5 Å$^{-2}$ for both O and D[78]. The O-D bond lengths were thermal-corrected using $B$, and upshifted further by 3% for the D-H conversion. The final O-H bond lengths in ice II are $(1.102 \pm 0.021)$ Å, $(1.064 \pm 0.021)$ Å, $(1.046 \pm 0.020)$ Å, and $(1.028 \pm 0.021)$ Å. The low frequency peak at 3194 cm$^{-1}$ is assigned to the local stretching of the longest O-H bond of 1.102 Å. The stretching vibration can be coupled both intra- and intermolecularly among the remaining three O-H bonds due to their close bond lengths, resulting in uncertainty in vibration frequency assignment. Hence, we excluded these three data points and only focused on the longest bond.

Ice VI contains two types of O$_4$ octahedra structure interpenetrating into each other, revealed by the neutron diffraction of D$_2$O ice[79]. For the first O$_4$ octahedra network, all proton positions are equivalent, yielding one O-D bond length $(0.986 \pm 0.048)$ Å, indicating a strong intramolecular coupling in stretching vibration. The second O$_4$ octahedra structure has three O-D bond lengths at $(0.937 \pm 0.055)$ Å, $(0.976 \pm 0.051)$ Å, and $(0.939 \pm 0.032)$ Å with multiplicity 1, 1, and 2, yielding an average of $(0.951 \pm 0.027)$ Å. These two bonds are thermal-corrected to $(0.996 \pm 0.048)$ Å and $(0.961 \pm 0.027)$ Å using $B = 2.99$ Å$^{-2}$ for oxygen and 3.80 for hydrogen[79], and further upshifted to $(1.026 \pm 0.049)$ Å and $(0.993 \pm 0.028)$ Å for the D-H conversion. These two bonds are associated with Raman stretching modes at 3390 and 3329 cm$^{-1}$[80].

For ice VIII, only one bond length is given at $(0.969 \pm 0.007)$ Å in D$_2$O ice at 10 K[81], which is thermal-corrected to $(0.981 \pm 0.007)$ Å using $B = 0.58$ Å$^{-2}$ for O and 1.54 for D, and further upshifted to $(1.010 \pm 0.009)$ Å because of D-H conversion. The O-H Raman vibration frequency is taken as the average of three intramolecular-coupled normal modes locating at 3477.4 cm$^{-1}$ ($v_1B_{1g}$), 3447.6 cm$^{-1}$ ($v_3E_g$) and 3358.8 cm$^{-1}$ ($v_1A_{1g}$) at 100 K[82], which is 3427.9 cm$^{-1}$.

Ice IX is a proton-ordered phase. Neutron diffraction data on D$_2$O ice shows that protons are preferably arranged in one configuration (referred to as 'major site'), with a nonzero occupancy on the other configuration (referred to as 'minor site'). The O-D lengths for major and minor configurations are $(0.976 \pm 0.003)$ Å and $(1.02 \pm 0.06)$ Å. Thermal-corrected length is reported for only the major site at $(0.983 \pm 0.004)$ Å[83]. This value is further upshifted to $(1.012 \pm 0.004)$ Å for D-H conversion. Because of the low occupancy, we exclude the minor configuration bond length and only use that of the major configuration. Two intense Raman peaks of O-H stretching are found at

3280.7 cm$^{-1}$ and 3160.4 cm$^{-1}$ at 100 K[84], with the latter assigned to the vibration of O-H bond on major sites.

For ice XVII, two O-H stretching Raman peaks at 3106.3 cm$^{-1}$ and 3231.8 cm$^{-1}$ are reported in ref. [85]. These are regarded as decoupled vibration because their separation is >100 cm$^{-1}$. Two O-D bond lengths are given by neutron diffraction on D$_2$O ice at (1.023 ± 0.007) Å and (1.006 ± 0.004) Å, which are thermal-corrected to (1.044 ± 0.007) Å and (1.018 ± 0.004) Å using $B = 2.37$ Å$^{-2}$ for O, 4.05 Å$^{-2}$ for D1 and 3.34 Å$^{-2}$ for D2, respectively[86]. These values are further upshifted to (1.075 ± 0.007) Å and (1.049 ± 0.004) Å for D-H conversion.

For CaSO$_4$·2H$_2$O (gypsum), the Raman frequency of O-H stretching is taken from this work, which is 3406 cm$^{-1}$ and 3484 cm$^{-1}$, measured at room temperature. The O-H bond lengths are taken from ref. [20], (0.961 ± 0.006) Å and (0.948 ± 0.004) Å at 320 K, which are thermal-corrected to (0.984 ± 0.006) Å and (0.966 ± 0.004) Å using $B = 1.749$ Å$^{-2}$ for O, 3.520 Å$^{-2}$ for D1 and 3.074 Å$^{-2}$ for D2, respectively. These values are further upshifted to (1.014 ± 0.006) Å and (0.995 ± 0.004) Å for D-H conversion.

For Li$_2$SO$_4$·H$_2$O, two thermal-corrected O-H bond lengths are (1.004 ± 0.003) Å and (0.997 ± 0.002) Å[87]. We associate these two bonds with two Raman peaks of O-H stretching at 3439 cm$^{-1}$ and 3480 cm$^{-1}$, respectively[88].

CuSO$_4$·5H$_2$O contains five water molecules per unit cell, among which four are coordinated to Cu$^{2+}$, and the remaining one is far away from Cu$^{2+}$ thus non-coordinated. To exclude the interactions with metal ions, we only focus on the non-coordinated water molecules in this study. CuSO$_4$·5H$_2$O exhibits multiple water stretching peaks, and the modes above 3300 cm$^{-1}$ is assigned to the stretching of non-coordinated water molecules, at 3360 cm$^{-1}$ and 3477 cm$^{-1}$[89,90]. The O-H bond lengths of the non-coordinated water molecules are (0.978 ± 0.005) Å and (0.936 ± 0.011) Å[91]. These values are corrected for thermal motion using anisotropic temperature parameters tensor ($\beta_{ij}$) to (1.005 ± 0.005) Å and (0.951 ± 0.011) Å. These two bonds are associated with 3360 cm$^{-1}$ and 3477 cm$^{-1}$ Raman peaks, respectively.

For Ba(ClO$_3$)$_2$·H$_2$O, a single thermal-corrected O-H bond length is reported at (0.958 ± 0.011) Å[92], suggesting an intramolecularly coupled vibration. Thus the vibration frequency is taken as the average between $\nu_1$ (3512 cm$^{-1}$) and $\nu_3$ (3582 cm$^{-1}$) vibration modes at 90 K[93], that is 3547 cm$^{-1}$.

For LiClO$_4$·3H$_2$O, the O-H bond length is taken as the average among three refinement series, (0.993 ± 0.004) Å, with uncertainty coming from the standard deviation of the three series[94]. The values provided in the reference have been thermal corrected during refinement. This bond length is associated with a sharp Raman peak in water stretching region, 3553 cm$^{-1}$[95].

For BaCl$_2$·2H$_2$O, four O-H bond lengths are given by neutron diffraction measurements at (0.953 ± 0.003) Å, (0.959 ± 0.003) Å, (0.960 ± 0.003) Å, and (0.972 ± 0.004) Å[96]. These values are thermal corrected to (1.0359 ± 0.003) Å, (1.0364 ± 0.003) Å, (1.0717 ± 0.003) Å and (1.077 ± 0.004) Å using $\beta_{ij}$. and assigned to four intermolecularly uncoupled Raman modes at 3456, 3352, 3318, and 3303 cm$^{-1}$[97].

The assignment and references are given in Supplementary Table 1, with all the data points plotted in Fig. 3b in the main text. The data points above were then fitted using a linear function with both X and Y errors, yielding a slope of (7.83 ± 1.26) × 10$^{-19}$ C and intercept of (0.957 ± 0.001) Å, corresponding to $\frac{\partial p}{\partial d}$ and d(0) in Eq. 3 in the main text. The uncertainty corresponds to 2$\sigma$. The reduced $\chi^2$ for this fitting is 4.96.

### Dipole moments vs O-H bond lengths in Fig. 3c

In the gas phase, equilibrium dipole moment of H$_2$O molecule is 1.855 ± 0.001 D, given by Stark effect measurements[29]. The H − O − H angle in gas phase is 104.48° and the O − H bond length is 0.9578 Å, as determined by fitting experimental rotational energy levels with the rotation-vibration Hamiltonian[26]. This gives a dipole moment of the O − H bond $p_{OH}$ = 1.514 ± 0.001 D.

Effective dipole moment of liquid water at ambient conditions is 2.9 ± 0.6 D, estimated from 0.5$e$ (±20%) charge transfer along each O − H bond, using synchrotron x-ray diffraction measurements[32]. The H − O − H angle and the O − H bond length are 103.924° and 0.984 Å, obtained from a joint refinement of X-ray and neutron diffraction data[98]. No iso- or anisotropic temperature parameters available, so no correction was made to the O-H bond length. This gives dipole moment of O-H bond $p_{OH}$ = 2.4 ± 0.5 D.

Ice Ih has a dipole moment of 3.09 ± 0.04 D, calculated using multipole iterative method, with error extracted from Fig. 1 in the ref. [33] The O-H bond length and H-O-H angle are 1.014 Å and 109.21° from neutron diffraction data at 60 K. The bond length is further thermal corrected to 1.0125 Å[75]. The $p_{OH}$ is calculated to be 2.67 ± 0.04 D.

### Polarizability and dielectric behavior of confined water

We consider a dielectric with $\varepsilon_r = \varepsilon/\varepsilon_0$ consisting of polarizable dipoles with the polarizability along the field direction $\alpha_{c,i}$ and the number density of dipoles $N_i$. The polarization under the electric field $E$ is $P = (\varepsilon_r - 1)\varepsilon_0 E$, which equals to the sum of the polarization from all dipoles, $P = \sum_i \alpha_{c,i} E N_i$, here $i$ denotes different types of dipoles. This leads to Eq. 8 in the main text, the dielectric constant of a confined water system.

From Eqs. 1, 2 and 6 in the main text, polarizability can be written in terms of total electric field $E$ along O-H bond:

$$\alpha(E) = \frac{\left(\frac{\partial p}{\partial d}\right)^2 k_0^2}{\left(k_0 - \frac{\partial^2 p}{\partial d^2} E\right)^3} \tag{14}$$

Taking gypsum as an example of nano-confined water system, the polarizability of the four types of O-H dipoles in gypsum are doubly degenerated and dictated by the local $E_{HB}$ field, 5.33 and 3.82 V nm$^{-1}$ for O-H$_A$ and O-H$_B$, respectively. In the presence of external electric field, the four dipoles have four different local fields, and thus the structure symmetry is broken, resulting in two polarizabilities splitting towards two directions (Supplementary Fig. 5c). By adding up the four field-dependent polarizabilities according to Eqs. 7 and 8 in the main text, dielectric constant $\varepsilon_r$ = 6. 6 ± 0.7 was obtained for water in gypsum. The calculated field-tunable $\varepsilon_r(E)$ is plotted in Supplementary Fig. 5d, which presents a modulation of <0.1% due to the field-tuning effect on confined water layers in gypsum.

To observe the field-tunable $\varepsilon_r(E)$ in gypsum, we performed capacitance measurements in one of our devices, using capacitance bridge (Supplementary Fig. 5a). In the measurement, a graphite/gypsum/graphite capacitor fabricated on SiO$_2$ (290 nm)/Si substrate was placed in vacuum (<10$^{-6}$ mbar). The capacitance of the device was measured using AH 2700 capacitance bridge with an ac excitation of 1 V at 1.2 kHz.

The equivalent circuit is sketched in Supplementary Fig. 5a. The experimental capacitance measured by the bridge is a combination of four capacitors with three contributions: capacitance of the gypsum ($C_g$), quantum capacitance of top and bottom graphite electrodes ($C_q$), and parasitic capacitance from the measurement setup ($C_p$). From capacitors model we can fit the experimental curve using:

$$C = C_p + \frac{1}{1/C_{q1} + 1/C_{q2} + 1/C_g} \tag{15}$$

Field-independent $C_p$ is a fitting parameter with constant value. $\varepsilon_r$, which is embedded in $C_g$, is another fitting parameter. $1/C_{q1}$ and $1/C_{q2}$ are function of carrier density on graphite electrodes and are given by

ref. 99, and we obtain $C_p \approx 319$ fF and $\varepsilon_r \approx 3.68$. With $C_p$ subtracted from the experimental curve, the modulation of capacitance is -0.9%, much higher than the 0.1% modulation expected for field-tunable polarizability of confined water. Thus, it is concluded that the effect of field-tunable polarizability of water molecules is hidden in the graphite quantum capacitance.

Gypsum is a HBH consisting of confined water layers and CaSO$_4$ layers, which can be modeled as two capacitors in series. The total $\varepsilon_r$ of gypsum has been determined above. The capacitance of confined water layers can be evaluated using number density of confined water molecules and Eq. 8 in the main text. Gypsum (CaSO$_4 \cdot$2H$_2$O) has molecular weight of 172.17 g·mol$^{-1}$ and mass density 2.32 g·cm$^{-3}$[100], yielding a number density of $8.11 \times 10^{27}$ m$^{-3}$ of gypsum, and $1.62 \times 10^{28}$ m$^{-3}$ of H$_2$O. Considering that the water molecules are confined between CaSO$_4$ layers, with confining space taking up 40% of the gypsum volume (confining length measured between O$_S$ and O$_S$', the oxygen atoms in SO$_4$ groups from adjacent layers), the number density of water molecules in the confining space is corrected to $4.06 \times 10^{28}$ m$^{-3}$. Equation 8 in the main text produces $\varepsilon_r = 6.6 \pm 0.7$ for confined 2D water layers in gypsum. Using capacitor-in-series model, $\varepsilon_r = 2.8 \pm 0.5$ can be obtained for CaSO$_4$ molecular sheets. This value matches well the reported $\varepsilon_r$ for anhydrite (CaSO$_4$), which is $2.45 \pm 0.52$ for γ-phase and $2.70 \pm 0.02$ for β-phase[43].

Our model finds that water sheets confined in gypsum presents a lower $\varepsilon_r$ compared to its bulk form due to the restricted rotational freedom. Such behavior is similar to water confined in carbon nanotubes (CNT). Simulations has shown that in CNT-confined water, axial component of $\varepsilon_r$ increases to above 100 with a relaxation time longer than bulk water, while the $\varepsilon_r$ of water normal to the carbon walls decreases drastically (to ~6), with the dipolar relaxation time severely suppressed[101,102]. The small normal component of $\varepsilon_r$ is attributed to the restricted dipole rotation, caused by the strong orientation of water molecules near walls due to induced surface charge[103]. A similar decrease in $\varepsilon_r$ was also found at the water−C$_{60}$ fullerenes interface, which is attributed to the electrostatic interaction between water and surface charge on C$_{60}$[104,105].

## Dielectric constant of liquid water

We consider the dielectric response of liquid water coming from two parts: dipole reorientation, as well as atomic and electronic polarization. The former is given by Langevin-Debye equation[106]:

$$\left\langle P_{dipole} \right\rangle = \frac{N_0 p^2 E}{3kT} \tag{16}$$

where $N_0$ and $p$ are the number density and dipole moment of the H$_2$O molecule, $k$ is Boltzmann constant, $T$ the temperature and $E$ is the electric field. The number density of H$_2$O molecule is $3.35 \times 10^{28}$ m$^{-3}$, using the mass density of liquid water. The latter, atomic and electronic polarization, is given by considering individual O-H dipole moments of number density $2N_0$ and dipole moment $p$, with orientation given by Boltzmann distribution of $p$ in external $E$-field. The induced dipole moment per O-H bond is $\alpha E(\cos\theta)^2$, where $\theta$ is the angle between $p$ and $E$ (defined in Fig. 1b, main text). Integration over all 3D space gives the polarization density of induced dipole:

$$P_{a+e} = 2\alpha E N_0 \frac{\int_1^{-1} y^2 e^{xy} dy}{\int_1^{-1} e^{xy} dy} \tag{17}$$

with $y = \cos\theta$, and $x = pE/kT$. Using Taylor expansion on the integral term and keeping to quadratic term, Eq. 17 reduces to

$$P_{a+e} = 2\alpha E N_0 \left[ \frac{1}{3} + \frac{1}{45}\left(\frac{pE}{kT}\right)^2 \right] \tag{18}$$

Adding up the two contributions and using $P = (\varepsilon_r - 1)\varepsilon_0 E$, we obtain dielectric constant of liquid water:

$$\varepsilon_r = 1 + \frac{N_0}{3\varepsilon_0}\left[ \frac{p^2}{kT} + 2\alpha + \frac{2\alpha}{15}\left(\frac{pE}{kT}\right)^2 \right] \tag{19}$$

where $p$ and $\alpha$ are be given by our model from Eqs. 6 and 7 in the main text.

Although our model is derived from individual water molecules confined in gypsum, it successfully reproduced the dipole moment of liquid water, 2.66 D, where water dipoles are free to rotate and interact with each other. Using this result and Eq. 7 in the main text, a static dielectric constant $\varepsilon_r = 27 \pm 2$ was obtained for liquid water. This is significantly lower than experimental value $\varepsilon_r = 79$ due to the complex hydrogen bonding network of water clusters in liquid water, which enhances the interaction between water molecules and thus alters the cooperativity of water molecules with HBs and affects the dielectric response. A conventional approach to include such correlation effect is to multiply $p^2/kT$ term in Eq. 19 with a dipole correlation factor $G_k$[44]. To get experimental $\varepsilon_r = 79$, $G_k$ of 3.15 should be taken, in the range of reported $G_k$, from 2.72 to 3.70 which depends on the different models used for the simulation[45]. With bond length taken as 0.984 Å[98], our model yields a dielectric constant value of $78.7 \pm 7.2$ at $E = 0$ V nm$^{-1}$ and $78.8 \pm 7.2$ at $E = 0.5$ V nm$^{-1}$, with the electronic polarization contributes to only 3% of the dipole reorientation.

From the calculations above, it is seen that in bulk water, dipole reorientation contributes over 97% to the dielectric response. Previous studies have shown that in confined liquid water, particularly when the confinement is less than 2 nm, the perpendicular dielectric constant drastically decreases to below 10[41,42,49]. This significant reduction is a result of restricted rotational freedom. To quantify the contribution of electronic and atomic polarization, which is highly anisotropic and dependent on the water molecule orientation, we calculated the dielectric constant resulting from this effect only. As illustrated in Supplementary Fig. 6, the dielectric constant has a maximum of 13 from non-reorientation polarization. Given the experimental $\varepsilon_r = 79$, dipole reorientation consistently contributes more than 84% of the dielectric response.

Consequently, in scenarios where confinement restricts dipole reorientation, and the dielectric response is entirely due to electronic and atomic polarization, water consistently exhibits a low $\varepsilon_r$.

## Water-carbon interface

In the case of single water molecules trapped inside a C$_{60}$ fullerene, both the symmetric $\nu_1$ and antisymmetric $\nu_3$ stretching modes are significantly red-shifted compared to free H$_2$O, to 3573 cm$^{-1}$ and 3659.6 cm$^{-1}$, respectively. In a harmonic approximation, this corresponds to a force constant $k \approx 725$ N m$^{-1}$. Using Eqs. 3 and 4 in the main text, we obtain a local $E$-field $E_{HB} = 1.7 \pm 0.2$ V nm$^{-1}$, O-H bond length $d_{OH} = 0.974 \pm 0.004$ Å, and dipole moment (using Eq. 6 in the main text) $p_{H2O} = 2.37 \pm 0.14$ D. With our dipole-in-$E$-field model, the calculated interaction strength $U_{HB} = 1.5 \pm 0.2$ kcal mol$^{-1}$ per bond ($66 \pm 9$ meV), which is comparable to typical weak HBs. Similar values are obtained for water confined in single-walled carbon nanotubes, with stretching frequency at 3569 and 3640 cm$^{-1}$, yielding $k \approx 720$ N m$^{-1}$[53]. However, this picture is likely complicated by the interplay between water-water intermolecular HBs and water-carbon π HBs.

At the water-graphene interface, water molecules exhibit a peak around 3600 cm$^{-1}$, which was assigned to the stretching vibration of the 'dangling' O-H dipole pointing towards the graphene film[54]. The reported vibration frequency of $\nu_{OH} = 3616$ cm$^{-1}$ for the dangling O-H dipole, using our approach, corresponds to $k = 725.1$ N m$^{-1}$, $E_{HB} = 1.7 \pm 0.2$ V nm$^{-1}$, $d_{OH} = 0.975 \pm 0.004$ Å, $p_{H2O} = 2.37 \pm 0.14$ D, and $U_{HB} = 1.54 \pm 0.20$ kcal mol$^{-1}$ per bond ($67 \pm 9$ meV). Our model predicts

the electric field exerted by the π electrons on the water O-H dipoles is around 1.7 V nm⁻¹. Using only the stretching frequency of confined water, our results directly reproduce key properties of confined water, including HB strength, local $E$-field, O-H bond length, and dipole moment.

This peak can also be tuned by applying an external electric field in the electric double layer region near graphene surface. In the vicinity of graphene, water O-H dipoles experience an $E$-field exerted from π electrons of carbon atoms. This interaction can also be considered a π-hydrogen bond (π-HB), which redshifts the stretching frequency of the O-H dipole compared to that of a free water molecule. In the presence of $E_{ext}$, the interaction is tuned, which can be seen in the shift of the stretching frequency of dangling O-H bond on the graphene surface.

Reference 54 reported a sum frequency vibrational spectroscopy (SFVS) study on the water-graphene interface. As an independent validation of our model, we used the data points from Fig. 4b in ref. 54. The vibration frequency of dangling O-H peak on y-axis is converted to force constant $k$ using harmonic approximation. The interfacial electric field $E_s$ on x-axis corresponds to the $E_{ext}$ in our study. $E_{tot}$ for each point is obtained using Eq. 5 in main text, with $E_{HB}$ given by Eq. 4 (main text) using $k$ at $E_{ext} = 0$, taken from Fig. 4b in ref. 54. Since the dangling O-H dipole points towards graphene film, cos ϑ in main text Eq. (5) is taken as 1. The obtained results match perfectly into our model, as plotted in Supplementary Fig. 7.

### Dielectric constants of water ice systems

Due to the presence of defects in proton-disordered ices which cause dipole reorientation and, thus, an anomalously large $\varepsilon_r$, we chose the proton-ordered ices to study the static $\varepsilon_r$. Hence, only atomic and electronic polarization contributions are considered. Ice XI is proton-ordered form of ice Ih, with $\varepsilon_r = 2.9$ taken from Fig. 6 in ref. 107. The authors cooled down the ordinary ice (ice Ih) and measured $\varepsilon_r$ at 77 K, where the ice Ih–XI phase transition is expected to occur. The observed small and non-dispersive $\varepsilon_r$ indicates a proton-ordered form of ordinary ice. The overall static $\varepsilon_r$ of proton-ordered ice II is around 4.2, measured from Fig. 6 in ref. 47, although a small increase was observed with frequency (not evaluated). Ice IX is proton-ordered form of ice III. The $\varepsilon_r$ of ice IX is taken as 4 according to Fig. 3 in ref. 108, where static $\varepsilon_r$ presented a sharp drop during the ice III–IX phase transition.

We examined the static dielectric constant along [001], [111] and [010] directions for these three phases of ices, respectively, axes direction defined in refs. 109–111. The three confined water systems possess 2, 4, and 6 types of non-equivalent O-H dipoles with respect to the assessing axis, with polarizability marked in Fig. 3d in main text. Using the neutron diffraction data of O-H bond lengths reported in refs. 75,78,112 (the bond length of ice III is thermal corrected to 1.005 Å following the approach in Methods, 'Thermal corrections for neutron structural data') and density reported in refs. 110,113, the dielectric constants calculated using Eq. 8 in the main text, are 3.8 ± 0.5, 5.9 ± 0.8, and 4.7 ± 0.4 for structure of ice XI, II and IX, respectively. Our results show reasonable agreement with the reported $\varepsilon_r$ for ice XI, II, and IX, which are 2.9[107], 4.2[47] and 4[108].

All the estimated $\varepsilon_r$ for water systems are summarized in Supplementary Table 2.

## Data availability

The Source Data files for all data presented in graphs within the Figures used in this study are available in the Zenodo database: https://doi.org/10.5281/zenodo.15019875.

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

## Acknowledgements

This research was supported by the European Research Council (ERC) under the European Union's Horizon 2020 research and innovation program (Grant Agreement No. 865590) and the Research Council UK [BB/X003736/1]. Q.Y. acknowledges the funding from Royal Society University Research Fellowship URF\R1\221096 and UK Research and Innovation Grant [EP/X017575/1].

## Author contributions

A.M. and Q.Y. initiated and supervised the project. Z.W. fabricated devices, performed Raman measurements and analyzed the data with the help from I.T. and A.N.G. A.B., M.Y., and F.P. performed DFT calculations. V.K. and A.N.G. helped with FTIR measurements. P.D.N performed scanning near-field microscopy measurements. C.M. carried out the low temperature capacitance measurements. T.T. and K.W. provided hBN crystals. A.M., Z.W., Q.Y., and K.S.N. contributed to writing the paper. All authors discussed the results and commented on the paper.

## Competing interests

The authors declare no competing interests.

## Additional information

**Supplementary information** The online version contains
supplementary material available at

Ziwei Wang, Qian Yang or Artem Mishchenko.

peer review file is available.

