## [Transparent Peer Review file · Nature Communications]

Quantifying hydrogen bonding using electrically tunable nanoconfined water

Corresponding Author: Professor Artem Mishchenko

Version 0:

Reviewer comments:

Reviewer #1

(Remarks to the Author)

In the manuscript, the authors propose a new approach to describe H-bonding as an elastic dipole in electric field, i.e., dipole-in-E-field approach. Parameters for the quantitative evaluation of H-bonding were obtained through spectroscopic measurements and analyses of previously reported experimental data, and the validity of this approach is discussed by evaluating various physical quantities of samples relating H-bonding. While this approach has been shown to be somewhat effective in predicting physical quantities, the reviewer has following concerns;

- 1) What generates the electric field E_{HB} in Fig 1a?(Especially in the condition of $E_{ext} = 0$.) Also, does its magnitude and direction depend on p or not?
- 2) Is it reasonable to apply the parameters obtained from gypsum measurements to other H-bonded systems such as water and ice? Considering the fact that the estimated H-bonding energies in P6L8 differ from the reported ones by about 10~30%, it seems not to be reasonable.

To my opinion, although there is no problem in applying this approach to water adsorbing inorganic salts, I think the authors need to be cautious about generalizing this, such as applying it to water or ice. Therefore, I could not recommend this paper for publication in Nature Communications, to my regret.

However, the band shift of the OH stretching vibration bands dependent on E_{ext} and the evaluation of H-bonding are interesting in itself, and I would like to suggest to re-summarize only the contents of inorganic salts and submit it to a journal in a specialized field.

Other comments are listed below.

Other Comments

P2 Fig. 1d Are the O atoms of water molecules in gypsum not involved in H-bonding as the proton acceptor? If so, the lack of consideration of H-bonding at the O atoms may be the reason for the poor accuracy for estimating the H-bonding energy. In previous studies on H-bonding of water, it has been suggested that not only proton donation, but also proton acceptance at the O atom can affect the spring constant (e.g., Phys. Chem. Chem. Phys. 7, 3005–3014 (2005)). I think the authors need to consider how the H-bonding on the O atom affects the E_{HB} and p .

P7 L3 from bottom "in the case of single water molecules trapped inside a C60 fullerene"

Since the negative charge inside C60 is delocalized and distributed on the inner surface, I think, it is inappropriate to consider in terms of the model like Fig. 1a.

Why not summarize all the values estimated in this study in a table and compare them with literature values to discuss their validity? It would make it easier for the reader to understand.

Reviewer #2

(Remarks to the Author)

The noteworthy result for this paper is that the dipole in E field approach can quantify the characteristics of hydrogen bonding in different environments. The authors do careful and in-depth work for their research, including +/- error for many equations, and provide a convincing and useful way of thinking. The paper also claims this is the first study of hydrogen

bond heterostructures. A Nature Comms paper was just published in July on relevant, but not identical materials: "Large-scale 2D heterostructures from hydrogen-bonded organic frameworks and graphene with distinct Dirac and flat bands." In contrast, this paper still takes the analysis in a more thorough vibrational spectroscopy direction than the other paper, which contains one in situ Raman spectrum in the supplemental. The more in-depth vibrational spectroscopy study coupled with simulations and neat equation analysis can give some useful property predictions and makes this work unique from the other recent publication.

The work could be made stronger by the following changes:

1. The authors write clearly and address some concerns/discrepancies in the paper, but ignore others. For example: they write: "For liquid water and ice Ih, our model predicts $\rho_{H_2O} = 2.66 \text{ D}$ and 3.35 D , respectively, which are close to the reported values of 2.9 D and 3.09 D ." This difference should then include a discussion and further analysis.
2. The authors do not address why $2e$ shows 300/80/10 for HA, but only 80k and 2 replicates of 300 for HB, but one is not complete. The HB data may have been too weak for 10? Unsure. And while the authors provide a clear explanation in their methods for why 300 K cannot be measured past $\pm 0.3 \text{ V/nm}$, they do not explain the missing 10 K data for -0.5 V for HA or missing 300 K rep 2 data for HB.
3. It is unclear why, if the authors have the spatial resolution available to them to measure these hydrogen bond heterostructures (high NA objectives for Raman and nano-FTIR using AFM), they do not show any spatially resolved data. Was spatially resolved data uninteresting? (no variations seen) or were all spatial locations averaged due to low SNR? Unknown from this paper. Related some experimental details are missing, such as Raman laser power.

Reviewer #3

(Remarks to the Author)

The authors present a highly interesting study on the spectroscopic properties of ultra-confined planar water films in gypsum from which they infer the water dielectric properties, using a combination of different spectroscopic methods, neutron scattering and theoretical modeling. While a few papers have claimed to measure reduction of the dielectric constant of water in confinement, this paper is probably the first to have accomplished this. I recommend publication but would like the authors to consider my comments to sharpen their interpretation and the connection to their theoretical modelling based on the Langevin equation.

The reduction of the water dielectric constant in confinement was predicted based on simulations in Schlaich et al. PRL 117, 048001 (2016), where it was also shown that the dielectric response becomes tensorial in confinement and that the perpendicular component is reduced only for confinement thickness below a nanometer and that the parallel component is pretty much unchanged. The tensorial character of the dielectric response might in fact be important for the interpretation of some of the reported spectroscopic results. The relation between polarization fluctuations and the dielectric constant is rather complex, and it is shown in that paper that the perpendicular dielectric properties are dominated by collective polarization effects, meaning that neighboring water dipoles tend to fluctuate in an anticorrelated fashion, quite in line with the spectroscopic findings in the paper. Importantly, the perpendicular collective contribution to the dielectric response in confinement is negative. This means that the dielectric response cannot be fully understood based on Langevin models for isolated dipoles. These results have been corroborated for very different surface chemistries in Loche et al. J. Phys. Chem. B 2020, 124, 4365–4371, where it was also shown that the effects seen for much larger confinement thicknesses in Ref. 45 are not contradictory to a dielectric constant that only changes for confinement thicknesses below about a nanometer. It would be good if the authors could specify in their paper precisely what the water slab thickness is (defined by the Gibbs surface separation) and which of the two tensorial dielectric components presumably controls the spectral shifts. A dichroic IR measurement, as done in Daldrop et al. NATURE COMMUNICATIONS | (2018)9:311 would be an excellent way to look into this, but I am not sure whether such a measurement would be feasible in the present setup.

Version 1:

Reviewer comments:

Reviewer #1

(Remarks to the Author)

The authors' replies were clear and almost satisfied the reviewer. However, I have not fully understood the following point on the previous question: 1) What generates the electric field E_{HB} in Fig 1a? (Especially in the condition of $E_{ext} = 0$.) Also, does its magnitude and direction depend on p or not?

In the authors' reply, the following explanations are included.

"Practically, the electric field p could affect the electronegativity of the acceptor, distorting the electron distribution on the acceptor and perturbing E_{HB} ".

"there is indeed a charge redistribution of the acceptor in the electric field of p ".

These explanations suggest that E_{HB} and p are not independent quantities but rather that they are dependent on each

other, which means that the eq (2) should be expressed as $U_{HB} = -p(E) \cdot E(p) + \dots$

Considering this equation, I have the following concerns.

1) Can E_{HB} simply be considered as the effective electric field generated by an electron acceptor? Can the contribution of p be ignored?

2) Does the effective electric field obtained from Raman measurements correspond to E in Figure 1 and eq (2)? Does E have the contribution of p ?

I think it is necessary to include discussions in the manuscript that addresses these concerns, since these questions are related to the basis of the model proposed by the authors.

Reviewer #2

(Remarks to the Author)

The authors have addressed the three reviewer's comments in this version of the manuscript. The manuscript and associated files appear ready for publication.

Reviewer #3

(Remarks to the Author)

The editors have answered all my questions in a satisfactory fashion, I can recommend publication of the paper as is.

Reply to Reviewer #1

In the manuscript, the authors propose a new approach to describe H-bonding as an elastic dipole in electric field, i.e., dipole-in-E-field approach. Parameters for the quantitative evaluation of H-bonding were obtained through spectroscopic measurements and analyses of previously reported experimental data, and the validity of this approach is discussed by evaluating various physical quantities of samples relating H-bonding. While this approach has been shown to be somewhat effective in predicting physical quantities, the reviewer has following concerns;

1) What generates the electric field E_{HB} in Fig 1a? (Especially in the condition of $E_{ext} = 0$.) Also, does its magnitude and direction depend on p or not?

We thank the Reviewer for the question, and we are sorry for the lack of clarity regarding this important point. As depicted in **Fig. 1a** in the manuscript, the \mathbf{E}_{HB} represents an effective field at the O-H dipole, \mathbf{p} , generated by the nearby electron acceptor A.

In a typical HB, the distance between acceptor and the O-H dipole is usually more than twice the length of the dipole, allowing us to treat this field as uniform at the dipole's position. Hence, the hydrogen bond in our work is conceptualised as the dipole \mathbf{p} placed in the uniform effective electric field \mathbf{E}_{HB} . The field \mathbf{E}_{HB} stretches or compresses the O-H bond due to the dipole-field interaction, changing the spring constant of O-H bond. Consequently, the behaviour of such hydrogen bonded water is different from free water molecules.

The influence of the electric field from \mathbf{p} on \mathbf{E}_{HB} has been embedded in our model. Practically, the electric field \mathbf{p} could affect the electronegativity of the acceptor, distorting the electron distribution on the acceptor and perturbing \mathbf{E}_{HB} . To quantify this effect, we calculate the effective \mathbf{E}_{HB} at the O-H dipole position according to Eq. 4 in the manuscript, using the Raman shift observed in O-H stretching. With the geometry of the HB known, we can then derive the partial charge on the acceptor O atom, which in our study ranges from 0.12e to 0.36e among different confined water systems. This relatively low partial charge on O atom suggests that there is indeed a charge redistribution of the acceptor in the electric field of \mathbf{p} . Since the \mathbf{E}_{HB} in our model is the effective electric field from the acceptor, where the dipole-field interaction has been taken into consideration, the perturbation from \mathbf{p} has been effectively integrated into our model, and we ensured that the influence of \mathbf{p} on \mathbf{E}_{HB} is not double-counted.

Following the Reviewer's comment, we have added detailed explanation on the origin of \mathbf{E}_{HB} in the revised manuscript.

2) Is it reasonable to apply the parameters obtained from gypsum measurements to other H-bonded systems such as water and ice? Considering the fact that the estimated H-bonding energies in P6L8 differ from the reported ones by about 10~30%, it seems not to be reasonable.

To my opinion, although there is no problem in applying this approach to water adsorbing inorganic salts, I think the authors need to be cautious about generalizing this, such as applying it to water or ice. Therefore, I could not recommend this paper for publication in Nature Communications, to my regret.

However, the band shift of the OH stretching vibration bands dependent on E_{ext} and the evaluation of H-bonding are interesting in itself, and I would like to suggest to re-summarize only the contents of inorganic salts and submit it to a journal in a specialized field.

We thank the Reviewer for this question. Our model is derived from the fitting shown in **Fig. 3b**, which incorporates data points from both inorganic solid hydrate salts and various phases of ice. As a result, the model is designed to be applicable to ice systems as well.

When applying our model to proton-disordered ice Ih, we do observe a discrepancy between our results and the reported values. Our model estimates the U_{HB} of 10.5 ± 1.2 kcal/mol, which is higher than the reported lattice energy of 7.04 kcal/mol, derived from calorimetric and spectroscopic data corrected for vibrational zero-point energy (S. Nanayakkara, J. Chem. Theory Comput. 2022, 18, 1, 562–579). Similarly, our model estimates the dipole moment of ice Ih as 3.39 ± 0.24 D, exceeding the reported value of 3.09 ± 0.03 D. Since we calculate U_{HB} using $U_{HB} = -\mathbf{p} \cdot \mathbf{E}$, a higher \mathbf{p} results in a greater U_{HB} value.

We attribute the difference in hydrogen-bonding energies to the ubiquitous defects in proton-disordered ice Ih. Proton hopping and formation of H_3O^+ and OH^- pairs in ice Ih weaken the hydrogen bonds. Ice has nineteen different phases, generally classified into proton-ordered or proton-disordered. In proton-ordered ice, hydrogen atoms occupy fixed positions and do not migrate within the crystal lattice. In contrast, proton-disordered ices contain vacancies and interstitial atoms that enable proton hopping among water molecules, leading to the formation of H_3O^+ and OH^- ions. This proton hopping mechanism also contributes to notably high dielectric constant of proton-disordered ice Ih, much higher than that expected for restricted dipole rotation.

Our model's estimate assumes a proton-ordered structure of ice Ih, with hydrogen atoms in fixed positions. However, ice Ih is a proton-disordered phase (Phys. Chem. Chem. Phys., 2015, 17, 12458), where proton hopping between lattice sites can generate H_3O^+ and OH^- pairs (Science, 1952, 115(2989): 385-390.), increasing the potential energy and weakening the hydrogen bonds. Our model does not account for this defect-induced weakening, which may lead to an overestimated dipole moment and hydrogen-bond energy.

To assess the impact of these defects, we compared our estimated dipole moment for ice Ih – which led to an overestimated U_{HB} – with that for ice XI, a proton-ordered phase of ice Ih. The dipole moment produced by our model, 3.39 ± 0.24 D, agrees well with the 3.3 D dipole moment for a water molecule in ice XI, as reported by DFT simulations (Ishii, F., 2011, Molecular Simulation, 38(5), 369–372). This agreement demonstrates the applicability of our model to proton-ordered ice structures, confirming its validity for such systems.

Our estimated HB energy (U_{HB}) for liquid water per HB, 5.1 ± 0.5 kcal/mol, aligns well with the reference value of 5.26 kcal/mol, for liquid water HB strength, derived from the enthalpy of vaporisation of water at 25°C (D. R. Lide, CRC handbook of chemistry and physics 85th edition). We apologise for not specifying the uncertainty of the estimated U_{HB} value. This has now been included in the revised manuscript.

Our model primarily addresses scenarios with confined water, where water-water interactions are limited. Although our model was developed based on isolated water molecules, this simple approach has proven effective in describing even more complex water systems, including proton-ordered ice which is free from defects that could alter electrostatic interactions, and liquid water where, water molecules form an extensive hydrogen bonding network with cooperative interactions.

Beyond water and ice systems, our model can be generalised to any water system where electrostatic interactions are dominant. In this study, we developed a universal method to probe the confining strength of electrostatically confined water, and specifically, the O-H dipole. This confinement includes conventional hydrogen bonding, where the electric field arises from adjacent anions; π -hydrogen bonding, where the field originates from nearby π -electron systems such as benzene or alkenes (Chem. Rev. 2005, 105, 10, 3513–3560; J. Phys. Chem. Lett. 2011, 2, 22, 2930–2933); and external confinement, where the electric field is applied by external sources like DC voltage.

To achieve this, we used gypsum as a platform, where water molecules are naturally confined within the crystal structure by hydrogen bonding. In our experiments, this confining environment was perturbed by an external electric field, similar in principle to hydrogen bonding. This perturbation allowed us to observe the dynamics within the intrinsic binding environment and assess the response of water molecules to other types of electrostatic confinement.

Importantly, we would like to emphasise that our approach moves beyond traditional concepts of chemical bonds, such as hydrogen bonds or π -bonds. Instead, we interpret all these interactions purely as electric fields, analogous to an external DC field. This perspective allows our model to be generalised to other systems, including liquid water and ice within a hydrogen bonding network, carbon-water interfaces involving π -electrons, and systems subjected to external electric fields. We believe that the level of generality achieved by our model warrants broader readership.

In the revised manuscript, we have added further details on generalising our model to water, ice, and other electrostatically confined systems. The overestimation of dipole moment and U_{HB} for ice Ih has also been addressed.

Other comments are listed below.

Other Comments:

P2 Fig. 1d Are the O atoms of water molecules in gypsum not involved in H-bonding as the proton acceptor? If so, the lack of consideration of H-bonding at the O atoms may be the reason for the poor accuracy for estimating the H-bonding energy. In previous studies on H-bonding of water, it has been suggested that not only proton donation, but also proton acceptance at the O atom can affect the spring constant (e.g., Phys. Chem. Chem. Phys. 7, 3005–3014 (2005)). I think the authors need to consider how the H-bonding on the O atom affects the E_{HB} and p .

We thank the Reviewer for highlighting this aspect, which indeed deserved more detailed discussion in our manuscript. Gypsum is a unique system where water molecules do not interact with each other but are only hydrogen bonded to the CaSO_4 molecular walls. Consequently, the oxygen atoms in water (denoted as O_w) do not serve as proton acceptors for other water molecules, indicating the absence of intermolecular HBs between them. This effectively isolates the water molecules from one another, treating each as an individual molecule confined by CaSO_4 walls, without any HBs within the water layer. This is supported by two key factors: 1) the distance between O_w and the nearest H atom from another water molecule is $>2.8 \text{ \AA}$, significantly greater than the typical distance observed in conventional H...O hydrogen bond, which is between $1.2\sim 2.1 \text{ \AA}$ (Acta Cryst. 1988, B44, 341-344) (For comparison, the hydrogen bonds analysed in this study exhibit an H...O bond length of $\sim 1.8 \text{ \AA}$, **Fig. 1d**). 2) Our DFT simulations confirm that there are no electrostatic interactions between water molecules in gypsum.

Due to the simple confining environment free from intermolecular O_w HBs within the water layer, we chose gypsum as an ideal platform to study HB confinement of individual water molecules to their environment. In gypsum, the oxygen atoms in the CaSO_4 confining wall (denoted as O_s) act as the only HB acceptors. Our model treats the O-H... O_s HB as an O-H dipole in the electric field of HB acceptor, O_s . The HB acceptor provides the electric field, and the bonding strength can be quantified from the dipole – E -field interaction. This approach treats HB as universal electrostatic force instead of conventional chemical bond, regardless of crystal structure and chemical composition. Therefore, our approach is applicable to any system dominated by electrostatic interactions, even those with multiple hydrogen acceptors. This allows for future expansion of our model to include systems with O_w -based HBs or even more complex hydrogen bonding networks, where the effect from the nearby O_w can be assessed using the principle of superposition.

The role of O_w in HBs within gypsum has been clarified in the revised manuscript, incorporating the references suggested by the Reviewer.

P7 L3 from bottom "in the case of single water molecules trapped inside a C60 fullerene". Since the negative charge inside C60 is delocalized and distributed on the inner surface, I think, it is inappropriate to consider in terms of the model like Fig. 1a.

We thank for the Reviewer's feedback. Our approach treats HB as universal electrostatic interactions, meaning that the HB acceptor is essentially an electric field acting on the O-H

dipole, regardless of its spatial distribution. This reinterpretation allows us to extend our approach to any electrostatic interaction between water and the environment, going beyond the conventional O-H...A hydrogen bonding scenarios. Consequently, while the delocalised charges on C₆₀ are not considered as conventional HB acceptors, its electrostatic characteristics still allow it to be effectively analysed and evaluated using our approach. Since the two O-H dipoles are not positioned exactly at the centre of the C₆₀ cage, the net field resulting from the charges on the C₆₀ surface is non-zero. In our model, these charges on C₆₀ surface are equivalent to a hydrogen bond acceptor since they influence the O-H dipoles in the same manner – by stretching the dipoles. This perspective also supports the validity of our model across various electrostatically governed systems, including liquid water, various forms of ice, and water molecules confined in C₆₀ structures. Consequently, we believe that our approach offers a robust method for evaluating interactions within these systems.

Why not summarize all the values estimated in this study in a table and compare them with literature values to discuss their validity? It would make it easier for the reader to understand.

We appreciate the Reviewer's suggestion. We have summarised and compared all our estimations of dipole moments and HB energy of gas, liquid water and ice Ih, where applicable, with literature values in **Extended Data Table 2** of the revised manuscript. The estimated dielectric constants of 2D water sheets and CaSO₄ in gypsum, ice IX, ice II and ice XI are also included. The HB energy of water confined in C₆₀ is also calculated from the Raman data reported.

Reply to Reviewer #2

The noteworthy result for this paper is that the dipole in E field approach can quantify the characteristics of hydrogen bonding in different environments. The authors do careful and in-depth work for their research, including +/- error for many equations, and provide a convincing and useful way of thinking. The paper also claims this is the first study of hydrogen bond heterostructures. A Nature Comms paper was just published in July on relevant, but not identical materials: "Large-scale 2D heterostructures from hydrogen-bonded organic frameworks and graphene with distinct Dirac and flat bands." In contrast, this paper still takes the analysis in a more thorough vibrational spectroscopy direction than the other paper, which contains one in situ Raman spectrum in the supplemental. The more in-depth vibrational spectroscopy study coupled with simulations and neat equation analysis can give some useful property predictions and makes this work unique from the other recent publication.

We are grateful to the Reviewer for acknowledging the novelty of our work and for directing our attention to an interesting study published in Nature Communications (Nat Commun 15, 5934 (2024)), where the intralayer HBs between monolayer graphene and hydrogen-bonded frameworks (HOF) were investigated. As we have mentioned in the conclusion of our manuscript, hydrogen bond heterostructures introduced in our study can be integrated with HOFs for future applications in gas separation, catalysis, and (opto)electronic devices. The study highlighted by the Reviewer further reinforces the potential of our approach. Our concept of hydrogen-bonded heterostructure is an analogue to van der Waals heterostructures, where the interlayer hydrogen bonding (rather than van der Waals) binds the layers together. The referenced study focuses on intralayer HBs within the HOF layer. In contrast, our research with gypsum emphasises the interlayer binding forces, where water sheets are bonded to CaSO₄ confining walls via interlayer HBs along the out-of-plane direction. Despite the different systems under investigation, the above-mentioned work offers a valuable perspective on the mechanisms of hydrogen bonding in 2D heterostructures. This reference has been cited in the revised manuscript, serving as an example to compare different 2D hydrogen-bonding systems.

The work could be made stronger by the following changes:

1. The authors write clearly and address some concerns/discrepancies in the paper, but ignore others. For example: they write: "For liquid water and ice Ih, our model predicts $\mu_{H_2O} = 2.66 D$ and $3.35 D$, respectively, which are close to the reported values of $2.9 D$ and $3.09 D$." This difference should then include a discussion and further analysis.

We thank the Reviewer for the comment. We apologise for not clarifying the uncertainty associated with the dipole moments. For liquid water, a dipole moment of $2.9 \pm 0.6 D$ was obtained from charge transfer calculations along each O-H bond using x-ray diffraction data (Y. S. Badyal *et. al.*, J. Chem. Phys. 112, 9206–9208, 2000). The dipole moment of liquid water predicted by our model, $2.66 \pm 0.13 D$, agrees well with the reported value within the experimental uncertainty range. This consistency is further corroborated by results from *ab*

initio cluster calculations, which estimate a dipole moment of 2.7 D (J. K. Gregory *et. al.*, Science 275,814-817, 1997), and molecular dynamics simulations, which suggest a dipole moment of 2.75 D (Liem X. Dang, J. Phys. Chem. B 1998, 102, 3, 620–624). The uncertainties have been updated in the revised manuscript.

Our model produces a dipole moment of ice Ih of 3.39 ± 0.24 D, higher than the reported 3.09 ± 0.03 D. The overestimated dipole moment produced by our model is due to the proton hopping phenomenon in ice Ih. The ice Ih is a proton-disordered phase containing defects that enable proton hopping among different H atom sites and the formation of H_3O^+ and OH^- ions (Bjerrum N., Science, 1952, 115(2989): 385-390). Our result, however, was obtained assuming a proton-ordered structure. To exclude this effect, instead of comparing our estimation with ice Ih, we compare our results with its proton-ordered counterpart, ice XI. The dipole moment produced by our model based on a proton-ordered structure, 3.39 ± 0.24 D, agrees well with that in ice XI, 3.3 D, given by DFT simulations. We have made a more detailed discussion about the dipole moment of ice Ih in the revised manuscript.

2. The authors do not address why 2e shows 300/80/10 for HA, but only 80k and 2 replicates of 300 for HB, but one is not complete. The HB data may have been too weak for 10? Unsure. And while the authors provide a clear explanation in their methods for why 300 K cannot be measured past +/-0.3 V/nm, they do not explain the missing 10 K data for -0.5 V for HA or missing 300 K rep 2 data for HB.

We thank the Reviewer for highlighting this issue. The field-dependent data for O-H_A and O-H_B were not collected simultaneously because the optimal stretching activity occurs at different orientations, as shown in **Fig. 1f**. At 10 K, the intensity of O-H_B stretching is too weak, even at the orientation with optimal stretching activity. This is due to the experimental setup required at 10 K, which uses a microscope objective with a longer working distance and a smaller numerical aperture, significantly reducing the amount of collected Raman scattering light. Consequently, the acquisition time for a single point is exceedingly long, increasing the risk of sample damage due to laser exposure, and potentially resulting in a very low signal-to-noise ratio for the collected spectra. This makes data collection and analysis for O-H_B stretching at 10 K particularly challenging. Besides, measurements at 10 K are intended only as a supplement to those conducted at higher temperatures. Therefore, we opted not to perform such measurements in our experiments.

The missing data points for 300 $\text{K}_{(2)}$ O-H_B is due to the degradation of top few-layer-graphene (FLG) electrode under external electric field. During measurements, FLG electrodes can degrade into amorphous carbon at the FLG/gypsum interface when subjected to high electric fields. The breakdown voltage at which it occurs depends on several factors, including the thickness of FLG, the presence of defects and bubbles at FLG/gypsum interface, and the duration the device is subjected to electric fields. For the 300 $\text{K}_{(2)}$ O-H_B stretching, only two data points were collected before the top FLG degraded. Despite the limited data, these points were included in our analysis for consistency.

The absence of data points at negative voltage polarity for 10 K O-H_A is due to several reasons. First, reversing the field polarity does not provide additional insights; it merely provides extra data points for fitting. As shown in **Fig. 1d**, due to the symmetry in the gypsum structure, a positive electric field acting on water molecules 1 and 2 has the same effect as a negative field on molecules 3 and 4. Second, to protect the top FLG electrode from potential degradation, we aim to minimize the duration of laser exposure. Additionally, the 10 K measurement complements those at higher temperatures, serving to verify if similar behaviour is observed at lower temperatures rather than to explore new phenomena. Thus, for the 10 K O-H_A stretching measurements, we limited our data collection to positive fields.

3. It is unclear why, if the authors have the spatial resolution available to them to measure these hydrogen bond heterostructures (high NA objectives for Raman and nano-FTIR using AFM), they do not show any spatially resolved data. Was spatially resolved data uninteresting? (no variations seen) or were all spatial locations averaged due to low SNR? Unknown from this paper. Related some experimental details are missing, such as Raman laser power.

We appreciate the Reviewer's question. The Raman signal from the 2D water sheets within gypsum is inherently weak, requiring prolonged acquisition times to obtain a clear spectrum, even at room temperature, and even more so at low temperatures. Consequently, spatial Raman mapping would be highly impractical, as it would significantly extend the overall measurement duration. Additionally, because the electric field is uniform across the spatial area of our sample, the external field's effect on confined water in gypsum is consistent throughout, making spatial mapping unnecessary. Thus, even if we aimed to capture spatial distribution, the impracticality of extended acquisition times would limit such an approach, and a single-point measurement in both Raman and nano-FTIR is sufficient to reveal the field-dependent behavior of confined water in our study. Details have been updated in the **'Methods – Vibrational Spectroscopy Measurements'** section of the revised manuscript to clarify this.

The laser power of Raman measurements was restricted to below 8 mW to prevent potential damage to the sample. Spectra were collected before and after 8 mW Raman measurements using 1 mW laser, and the comparison of these spectra confirmed that the sample did not degrade during the measurements. Details on laser power, wavelength, and grating have now been included in the revised manuscript in the **'Method – Vibrational spectroscopy measurements'** section.

Reply to Reviewer #3

The authors present a highly interesting study on the spectroscopic properties of ultra-confined planar water films in gypsum from which they infer the water dielectric properties, using a combination of different spectroscopic methods, neutron scattering and theoretical modeling. While a few papers have claimed to measure reduction of the dielectric constant of water in confinement, this paper is probably the first to have accomplished this. I recommend publication but would like the authors to consider my comments to sharpen their interpretation and the connection to their theoretical modelling based on the Langevin equation.

The reduction of the water dielectric constant in confinement was predicted based on simulations in Schlaich et al. PRL 117, 048001 (2016), where it was also shown that the dielectric response becomes tensorial in confinement and that the perpendicular component is reduced only for confinement thickness below a nanometer and that the parallel component is pretty much unchanged. The tensorial character of the dielectric response might in fact be important for the interpretation of some of the reported spectroscopic results. The relation between polarization fluctuations and the dielectric constant is rather complex, and it is shown in that paper that the perpendicular dielectric properties are dominated by collective polarization effects, meaning that neighboring water dipoles tend to fluctuate in an anticorrelated fashion, quite in line with the spectroscopic findings in the paper. Importantly, the perpendicular collective contribution to the dielectric response in confinement is negative. This means that the dielectric response cannot be fully understood based on Langevin models for isolated dipoles. These results have been corroborated for very different surface chemistries in Loche et al. J. Phys. Chem. B 2020, 124, 4365–4371, where it was also shown that the effects seen for much larger confinement thicknesses in Ref. 45 are not contradictory to a dielectric constant that only changes for confinement thicknesses below about a nanometer. It would be good if the authors could specify in their paper precisely what the water slab thickness is (defined by the Gibbs surface separation) and which of the two tensorial dielectric components presumably controls the spectral shifts. A dichroic IR measurement, as done in Daldrop et al. NATURE COMMUNICATIONS | (2018)9:311 would be an excellent way to look into this, but I am not sure whether such a measurement would be feasible in the present setup.

We appreciate the Reviewer for the very positive assessment of our work. The crystalline water confined in gypsum provides an excellent platform for us to study the electrostatic character of hydrogen bonds on water molecules. In gypsum, a two-dimensional water sheet, with a thickness of ~0.3 nm, can be considered as two-dimensional crystalline dipoles, without any hydrogen bonds linking them to each other. The absence of HBs between water molecules is supported by our DFT calculations. The only interaction on the water dipoles is the electrostatic field imposed by the surrounding CaSO₄ walls (hydrogen bond between the water O-H and a sulphate's oxygen atom). This is quite different from liquid water where a complex hydrogen bonding network connects water molecules together, forming water clusters with varying coordination numbers for each molecule. The absence of HBs within the 2D water layer in gypsum effectively isolates water molecules from each other, allowing O-H

dipoles to interact only with the CaSO₄ walls, which simplifies the analysis of electrostatic effects on the water molecules from their confining environments.

Additionally, gypsum also provides a fixed and stable confinement geometry for water molecules. In this environment, removing or inserting water molecules into the structure, as well as rotating the confined water dipoles, would require substantial energy. Consequently, the water molecules adopt highly directional, fixed positions and orientations within the electrostatic field created by the surrounding CaSO₄ walls. This behaviour contrasts with that of liquid water, where molecules have rotational freedom within the hydrogen-bond network, and their collective orientation follows a Boltzmann distribution.

In our study, crystalline water in gypsum is conceptualised as isolated dipoles that are only electrostatically confined to the CaSO₄ molecular walls with a fixed orientation. Consequently, we did not apply the Langevin model to the crystalline water in gypsum as we did for liquid water, because dipole fluctuations are not relevant in this fully confined context. In our model, the interaction strength between water and the CaSO₄ walls is equivalent to the energy of hydrogen bond in conventional terms. The electrostatic field from the CaSO₄ confining walls acts on the water dipoles, stretching the O-H bond and altering its vibrational frequency, which we detect through spectroscopic measurements.

This confined environment can be perturbed by an external electric field \mathbf{E}_{ext} , further modifying the O-H bond length and shifting the vibrational frequency, as measured in our experiments. As illustrated in **Fig. 1d** of our manuscript, the O-H dipoles with inversion symmetry share the same crystalline environment and vibration frequencies at $\mathbf{E}_{\text{ext}} = 0$. When an \mathbf{E}_{ext} is applied along b-axis, half of these dipoles stretch, while their inversion-symmetric counterpart compress. This different response results in a red- and blue-shift of stretching frequency respectively, leading to a peak splitting on O-H_A and O-H_B stretching spectrum. By measuring positions of these split spectral lines as a function of external electric field, we establish a quantitative relation between the O-H stretching frequency and confining strength. This model enables us to derive further details about hydrogen bonding, such as O-H dipole moment and HB energy, using parameters like O-H vibration frequency or O-H bond length.

To test the applicability of our model across different systems, we calculated the dipole moment of water molecules in liquid water as 2.66 D, based on the reported O-H bond length of 0.984 Å (A K Soper 2007 J. Phys.: Condens. Matter, 19, 335206). When determining the dielectric constant of liquid water, we accounted for dipole fluctuations, as water dipoles in the liquid state are not rigidly confined and can rotate within the hydrogen bonding network. Under the electric field, the orientation of water dipoles follows the Boltzmann distribution, and the overall dielectric response is described by the Langevin-Debye equation. Using our model's dipole moment of 2.66 D, we obtained a dielectric constant of 78.7 ± 7.2 , applying a dipole correlation factor $G_k = 3.15$. Additional details on the dielectric constant calculations for liquid water are provided in the '**Methods, Dielectric constant of liquid water**' section of our paper.

The Langevin model, commonly used for liquid water, also applies to confined liquid water (Schlaich et al., PRL 117, 048001, 2016; Loche et al. J. Phys. Chem. B 2020, 124, 4365–4371). However, it is not suitable for the 2D water sheets in gypsum, where water molecules behave as isolated dipoles with fixed orientation, lacking the rotational freedom typical in fluid environments. Despite this, both confined liquid water and 2D water sheets in gypsum exhibit similarly low dielectric constants in the direction perpendicular to the confining surface, compared to liquid bulk water.

In the manuscript, we evaluated the dielectric constant of a 2D water layer within gypsum. In this setup, the two-dimensional water bilayer between CaSO_4 walls, with a thickness of ~ 0.3 nm, shows a reduced perpendicular dielectric constant of 6.6 ± 0.7 , drastically lower than the bulk water value of ~ 79 . Such low dielectric constant is also seen in surface-confined liquid water (Schlaich et al., PRL 117, 048001, 2016; Loche et al. J. Phys. Chem. B 2020, 124, 4365–4371), where the in-plane parallel dielectric response is similar to the bulk value, but the perpendicular response significantly decreases when confinement is less than 1 nm and saturates at around 10. The low dielectric constant is a result of restricted rotational freedom. In gypsum, the water molecules are fixed in both position and orientation, thus their rotation and reorientation do not contribute to the dielectric response, and the dielectric constant is a result of electronic/atomic polarisation only. A similar situation is observed in surface-confined liquid water, where water molecules near the confining wall are restricted in out-of-plane rotation. Consequently, the predominant contribution to the perpendicular dielectric response in confined liquid water is also from electronic/atomic polarisation. However, these surface-confined water molecules retain rotational freedom in the in-plane direction, which is why the parallel dielectric constant remains comparable to that of bulk liquid water. The discussion and comparison between water in gypsum and confined liquid water have been updated in the revised manuscript.

To evaluate the role of dipole reorientation in the dielectric response, we analysed the contributions from electronic/atomic polarisation and compared them with those from dipole reorientation polarisation. The contribution from electronic/atomic polarisation is associated with the electron distribution within the O-H bond and the stretching direction of O-H dipole. This is characterised by the polarizability tensor, which in our model is simplified to a diagonal form, as we only consider contributions from the parallel response along the O-H bond. Thus, electronic/atomic polarisation is highly dependent on the orientation of water molecules: polarisation is maximised when the electric field aligns with the O-H bond and minimised when it is perpendicular. In our manuscript (page 8), we quantified this anisotropic contribution for water, based on the individual dipole model, using the density of confined water, 3.39 g/cm^3 (T. Dufils et. al. Chem. Sci., 2024, 15, 516-527). As shown in **Extended Data Fig. 6**, when considering only electronic/atomic polarisation, the resulting dielectric constant ϵ_r of confined water varies from 1 to 13, depending on the orientation of water molecules. This value could further decrease at lower water densities, reaching a minimum of 1.

The dipole reorientation polarisation is derived using the Debye-Langevin equation, as detailed in the '**Methods, Dielectric constant of liquid water**'. Considering only the dipole reorientation contribution, the resulting dielectric constant ϵ_r ranges from 66 to 89 using the reported $G_k = 2.72$ to 3.70 (H Yu et al. J. Chem. Phys. 118, 221–234, 2003). If we use the experimental $\epsilon_r = 79$, the contribution from electronic/atomic polarisation accounts for less than 16% of the total dielectric response. This analysis indicates that when rotational freedom is restricted, as in confined systems, and the dielectric response arises solely from electronic and atomic polarisation, the dielectric constant for water always decreases along the confining direction. The exact value of ϵ_r depends on the density and arrangement of water molecules, which in turn relates to molecular relaxation in various confined environments.

We also compared our results with those in Ref. 46 (L. Fumagalli et al., Science, 360, 1339-1342, 2018), which reported a reduced dielectric constant of 2.1 for liquid water confined to below 2 nm. In contrast, the 2D water layer in gypsum of ~ 0.3 nm, has a dielectric constant of 6.6. This difference in dielectric constants for confined water is attributed to the arrangements of water molecules under different confining environments. In our study, water molecules within gypsum are arranged such that the plane of the H₂O molecule is almost perpendicular to the confining surface, as illustrated in our **Fig. 1d**. In contrast, the study by L. Fumagalli et al. describes water molecules whose planes are nearly parallel to the confining surface, with only a slight angle between them (T. Dufils et al., Chem. Sci., 2024, 15, 516-527). To further validate this, we applied our model to predict the arrangement of water molecules reported by Fumagalli et al., yielding a dielectric constant of 2.1. The results, marked by the white lines in **Extended Data Fig. 6** in our manuscript, align with simulations by Dufils et al, which show that the H₂O plane is nearly parallel to the confining surface.

The ϵ_r , discussed above and investigated in our work refers to the static dielectric constant, which describes the dielectric response to a low-frequency electric field, allowing sufficient time for dipole reorientation. We did not explore the high-frequency dielectric constant, ϵ_∞ , which governs spectroscopic line shifts, as it is beyond the scope of this study. The spectral line splitting observed in our experiments arises from changes in the local confining environment of the O-H dipoles due to an external static electric field, which stretches or compresses the O-H bond, thereby shifting its vibrational frequency. Nevertheless, the behaviour of ϵ_∞ is also intriguing for understanding the dynamics of confined water, especially in gypsum where water can be modelled as a two-dimensional crystalline dipole. We appreciate the Reviewer's suggestion to consider potential measurements for future research directions. The dynamics can be studied using techniques like sum-frequency generation, surface-enhanced Raman spectroscopy, and dichroic infrared spectroscopy, which could offer valuable insights into the dynamics of confined water.

Following the Reviewer's suggestion, we have expanded the discussion in the '**Methods, Dielectric constant of liquid water**' section of our manuscript to provide a more detailed comparison between our model for isolated water molecules confined in gypsum and the Langevin model for liquid water, where a hydrogen-bonding network is present. We have

incorporated the three references suggested by the Reviewer to enhance and support our comparative analysis.

Reply to Reviewer #1

Reviewer #1 (Remarks to the Author):

The authors' replies were clear and almost satisfied the reviewer. However, I have not fully understood the following point on the previous question: 1) What generates the electric field E_{HB} in Fig 1a? (Especially in the condition of $E_{ext} = 0$.) Also, does its magnitude and direction depend on p or not?

In the authors' reply, the following explanations are included.

"Practically, the electric field p could affect the electronegativity of the acceptor, distorting the electron distribution on the acceptor and perturbing E_{HB} ".

"there is indeed a charge redistribution of the acceptor in the electric field of p ".

These explanations suggest that E_{HB} and p are not independent quantities but rather that they are dependent on each other, which means that the eq (2) should be expressed as $U_{HB} = -p(E) \cdot E(p) + \dots$

Considering this equation, I have the following concerns.

1) Can E_{HB} simply be considered as the effective electric field generated by an electron acceptor? Can the contribution of p be ignored?

We appreciate the Reviewer's insightful question. In our model, \mathbf{E}_{HB} represents the effective electric field generated by the acceptor A at the dipole's position. Importantly, \mathbf{E}_{HB} inherently includes the influence of the dipole (\mathbf{p}) due to the charge redistribution induced by their interaction. This dynamic interaction self-consistently modifies both the dipole moment \mathbf{p} and the electric field \mathbf{E}_{HB} .

To clarify, when a non-interacting dipole \mathbf{p}_0 approaches the acceptor, the interaction causes charge redistribution within both the dipole and the acceptor. This leads to a transition from the initial field \mathbf{E}_0 (in the absence of interaction) to the effective field \mathbf{E}_{HB} , and a corresponding change in the dipole moment from \mathbf{p}_0 (non-interacting) to $\mathbf{p}(\mathbf{E})$, as depicted in **Fig. R1**. As the dipole moves within the acceptor's field, \mathbf{E}_{HB} dynamically evolves, further influencing $\mathbf{p}(\mathbf{E})$. The final values of \mathbf{E}_{HB} and $\mathbf{p}(\mathbf{E})$ reflect this charge redistribution due to dipole-field interaction, directly determining the hydrogen bonding strength according to:

$$U_{HB} = -\mathbf{p}(\mathbf{E}) \cdot \mathbf{E}_{HB} \quad (\text{R1})$$

By treating \mathbf{E}_{HB} as the effective field, our model inherently accounts for charge redistribution and dipole-field interactions, eliminating the need to evaluate these effects separately.

Thus, \mathbf{E}_{HB} can indeed be considered the effective electric field generated by the acceptor, modified by the presence of the dipole. The contribution of \mathbf{p} is embedded in its influence on the effective field. To provide further clarity, we have expanded the **Methods** section ("Full form of the dipole-in-E-field approximation") with a detailed discussion on the dipole-in-field approximation.

Fig. R1, Illustration of dipole-acceptor interaction. **a**, When the dipole \mathbf{p} is far from the acceptor A, the local field from A is negligible at the dipole position, resulting in no interaction between the dipole and the acceptor's field. The free dipole moment is \mathbf{p}_0 , and the electric field generated by A is \mathbf{E}_0 . The blue arrow represents the dipole, and the green background indicates the strength of the acceptor's field. **b**, When dipole moves closer to A, the interaction between them effectively stretches the dipole and induces charge distribution in both the dipole and the acceptor. In this case, the dipole moment changes from \mathbf{p}_0 to effective dipole moment $\mathbf{p}(\mathbf{E})$, and the electric field generated by A changes from \mathbf{E}_0 to effective field \mathbf{E}_{HB} . The HB strength is calculated as the product of $\mathbf{p}(\mathbf{E})$ and \mathbf{E}_{HB} .

2) Does the effective electric field obtained from Raman measurements correspond to E in Figure 1 and eq (2)? Does E have the contribution of p ?

I think it is necessary to include discussions in the manuscript that addresses these concerns, since these questions are related to the basis of the model proposed by the authors.

We thank the Reviewer for this question. The effective electric field obtained from Raman measurements indeed corresponds to the \mathbf{E} in **Eq. (2)**. In our model, \mathbf{E} includes the intrinsic field provided by the acceptor (\mathbf{E}_{HB}) and, when applicable, the contribution from the external field (\mathbf{E}_{ext}) projected along the O-H bond direction. This is described by **Eq. (5)** in the main text:

$$E_{\text{tot}} = E_{\text{HB}} + E_{\text{ext}} \cos \theta,$$

where θ is the angle between the O-H bond and the external field.

Importantly, \mathbf{E}_{HB} inherently incorporates the contribution of \mathbf{p} due to the dynamic interaction between the dipole and the acceptor. As the dipole interacts with the acceptor, this interaction modifies the charge distribution, altering both \mathbf{E}_{HB} and \mathbf{p} . These effects are fully embedded in

our model, ensuring that the measured effective field already reflects the contribution from the dipole.

Therefore, while the contribution of \mathbf{p} to \mathbf{E}_{HB} is not explicitly separated, it is implicitly accounted for within our model. This approach provides a consistent framework for predicting hydrogen bond properties without requiring additional parameters. To clarify this further, we have expanded the discussion in the **Methods** section under "Full form of the dipole-in-E-field approximation."

Reviewer #2 (Remarks to the Author):

The authors have addressed the three reviewer's comments in this version of the manuscript. The manuscript and associated files appear ready for publication.

We thank the Reviewer for the highly positive feedback. We appreciate the work and contributions from the Reviewer during the revision process to improve the quality of the manuscript.

Reviewer #3 (Remarks to the Author):

The editors have answered all my questions in a satisfactory fashion, I can recommend publication of the paper as is.

We sincerely appreciate the Reviewer's positive feedback and thoughtful remarks. We thank the Reviewer for all the comments and suggestions regarding this work to enhance the manuscript to a higher level.